# Recognition of common shortwave protocols and their subcarrier modulations based on multi-scale convolutional GRU

**Jiuxiao Cao**[ID]☯, **Rui Zhu**☯*, **Zhen Wang**☯, **Jun Wang**☯, **Guohao Shi**☯, **Peng Chu**☯

School of Electronic Information, Xijing University, Xi'an, Shaanxi, China

☯ These authors contributed equally to this work.
* zhu_r10@163.com

**Data availability statement:** All relevant data are within the manuscript and its Supporting Information files.

## Abstract

Shortwave communication plays a vital role in disaster relief and remote communications due to its long-range capabilities and resilience to interference. However, challenges such as multipath propagation, frequency-selective fading, and low signal-to-noise ratios (SNR) significantly hinder automatic protocol and modulation recognition. Traditional signal processing approaches often fail under such conditions, whereas deep learning offers new possibilities for robust signal classification. This study proposes a Multi-Scale Convolutional GRU (MSC-GRU) model for the automatic recognition of three representative shortwave communication protocols—CLOVER-2000, 2GALE, and 3GALE—and their twelve subcarrier modulation formats. The model transforms temporal signals into two-dimensional representations, applies parallel convolutional branches with different receptive fields, and captures temporal dependencies through a bidirectional GRU. This hybrid architecture enhances both spatial feature diversity and sequential learning capacity. The dataset includes 45,000 labeled samples from both simulated and USRP-based real-world sources, evaluated using five-fold cross-validation. Results show that the MSC-GRU model achieves 100% recognition accuracy for protocol identification at SNR < -10 dB, and 80% accuracy for subcarrier classification at SNR < -8 dB. Standard deviations across folds are reported to ensure statistical reliability. With an inference time under 10 milliseconds per signal on a standard GPU, the model demonstrates practical feasibility for real-time deployment. These results confirm that MSC-GRU provides a robust and scalable solution for shortwave communication protocol recognition in complex environments.

## Introduction

Although shortwave communication has gradually been replaced by other technologies in modern communication systems, it remains indispensable in certain specialized fields [1]. Shortwave communication is critical in emergency scenarios and disaster relief efforts. When

**Funding:** Shaanxi Province Key Research and Development Program under Grant 2024GX-YBXM-114.

**Competing interests:** The authors have declared that no competing interests exist.

conventional communication methods fail, shortwave can still provide reliable long-distance transmission [2].

However, shortwave communication suffers from several inherent limitations. Its transmission quality is highly susceptible to natural conditions, such as solar activity and weather variations, as well as human-made interferences, such as electromagnetic disturbances and frequency congestion. These factors result in unstable communication reliability [3]. Furthermore, the finite spectrum resources limit its efficiency. As the demand for communication increases, these challenges become more pronounced [4].

To address the shortcomings of shortwave communication, recent advancements in computing and communication technologies have fostered the development of adaptive techniques [5]. Adaptive technologies can dynamically adjust transmission parameters, including modulation type, bandwidth, and power, in response to real-time channel conditions. This adaptability enhances spectral efficiency, improves signal stability, and increases resilience to interference [6]. The application of adaptive technologies not only improves communication quality but also revitalizes the use of shortwave communication.

With the rise of adaptive technologies, the forms of shortwave signals have become more diverse. Traditional modulation schemes are being replaced by more complex multi-modulation schemes and spread spectrum techniques [7]. This trend poses new challenges for spectrum monitoring. Traditional spectrum analysis methods are often inadequate for distinguishing complex signal types, especially in low signal-to-noise ratio (SNR) environments. The ability to accurately classify and differentiate between various shortwave signals has become a critical area of research.

Recently, deep learning (DL) and machine learning (ML) techniques have gained prominence in shortwave signal recognition. For example, Peng et al. [8] provides an overview of DL applications, such as convolutional neural networks (CNN), recurrent neural networks, and hybrid networks in modulation recognition. It highlights the advantages of different models across various signal environments, emphasizing how combining models can enhance recognition accuracy. Abd-Elaziz et al. [9] focuses on applying CNN within cognitive radio networks, demonstrating how CNN architecture improvements lead to higher classification accuracy in complex modulation signals. Similarly, Mohsen et al. [10] proposes an automatic modulation classification model using CNN, which efficiently classifies modulation signals.

Deep learning architectures, such as deep belief networks and multi-user deep learning models, have also been explored for signal recognition. For instance, Zhang et al. [11] discusses DBNs and their ability to extract features layer by layer for modulation recognition. Jajoo and Singh [12] tackles multi-user signal overlap issues by employing CNN to distinguish overlapping modulation signals. Njoku et al. [13] introduces a hybrid model combining CNN and gated recurrent units (GRUs) to improve robustness in noisy environments. Furthermore, Xiao et al. [14] provides an in-depth review of DL-based modulation classification algorithms, focusing on small dataset scenarios and strategies to enhance performance in limited data conditions.

Recent studies on free-space optical (FSO) systems have highlighted the importance of designing signal recognition mechanisms that remain robust under complex and noisy channel conditions. For instance, Elsayed et al. have demonstrated improvements in FSO-PON and UAV-FSO systems by using advanced modulation schemes (e.g., hybrid DPPM-PAPM and OFDM) and turbulence mitigation techniques to maintain high fidelity in signal detection under atmospheric impairments [15–17].

Although these works focus on optical domains, they share a common goal with this study: ensuring reliable signal recognition under severe channel distortions. Inspired by these advancements, our work addresses the shortwave wireless communication domain, where

signal fading, interference, and frequency-selective distortion pose significant challenges. By introducing a multi-scale convolutional GRU architecture, we aim to provide similar noise-resilient recognition capabilities tailored to the shortwave modulation environment.

Shortwave communication plays a critical role in long-distance transmission, disaster recovery, and remote area communications due to its strong resistance to environmental interference and the ability to propagate over large distances using ionospheric reflection. However, automatic recognition of shortwave communication protocols and modulations remains a significant challenge due to the complex nature of shortwave channels, which often involve multipath propagation, strong noise interference, and frequency-selective fading.

Recent advancements have demonstrated that deep learning (DL) techniques can automatically extract robust features from large-scale datasets, build efficient models, and significantly improve the accuracy of signal recognition tasks. DL methods, particularly convolutional neural networks (CNNs) and recurrent neural networks (RNNs) like gated recurrent units (GRUs), have been widely explored for general modulation recognition problems. CNNs excel at spatial feature extraction, capturing local structures within signals, whereas GRUs are proficient at modeling temporal dependencies across sequential data.

Nevertheless, the unique characteristics of shortwave communication—such as low signal-to-noise ratio (SNR), dynamic multipath effects, and rapidly changing channel conditions—introduce distinct challenges that standard DL approaches designed for general wireless channels may not adequately address. Effective solutions for shortwave environments require specialized signal processing and machine learning architectures that can simultaneously model spatial variations and time-sequential dynamics under severe noise and fading conditions.

To address these challenges, this study proposes a Multi-Scale Convolutional GRU (MSC-GRU) hybrid model. The MSC-GRU architecture is designed to extract diverse spatial features using parallel convolutional branches with multi-scale receptive fields and to capture temporal correlations through a bidirectional GRU module. This combination allows the model to fully exploit both spatial and temporal information in shortwave signals, significantly improving recognition robustness in complex channel environments.

Compared to conventional CNN-based models, the proposed MSC-GRU approach demonstrates superior performance by achieving higher recognition accuracy, better stability across different SNR levels, and faster training convergence. Furthermore, experimental results based on both simulated datasets and real-world USRP-collected signals validate the model's effectiveness, confirming its practical feasibility for near real-time shortwave protocol and modulation recognition.

1. MSC-GRU architecture: By incorporating multi-scale convolution layers for feature extraction and GRUs for capturing temporal dependencies, the proposed model enhances the recognition of complex shortwave signals with improved robustness.
2. Modulation scheme classification: The model is validated on three representative shortwave modulation schemes: 2GALE, 3GALE, and CLOVER-2000. Experimental results demonstrate its effectiveness under various channel conditions.
3. Subcarrier modulation identification: Beyond modulation classification, the model's capability is extended to recognize subcarrier modulation schemes, including twelve distinct types, achieving excellent performance even in low SNR environments.

## Ralated work

In this section, we provide an overview of several deep learning algorithms related to shortwave modulation recognition, with a focus on approaches involving image and convolution

processing, multi-scale methods, and temporal processing techniques. We analyze their roles in enhancing the accuracy of signal modulation classification, especially under complex signal environments. Through a review of relevant literature, we emphasize the advantages of each technology and its potential in addressing the challenges of recognizing modulation schemes in noisy and dynamic communication channels.

## Section A: Image or convolution processing in modulation recognition

CNN are widely used in image processing and signal classification due to their capability to efficiently extract local features through convolutional kernels. This approach not only effectively reduces model parameters and computational complexity but also handles structured data well [18–21]. For instance, CNN can treat the temporal and frequency characteristics of signal data as two-dimensional matrices. By transforming sequences into 2D representations, CNN become a popular strategy for modulation recognition, allowing them to capture spectrum features and modulation patterns more effectively, thereby improving classification accuracy. This is also the approach utilized in this paper, where segments of the signal sequence are transformed into 2D representations for subsequent feature extraction.

A key advantage of CNN lies in their parameter-sharing mechanism and local connectivity, which enables efficient processing of large input data volumes [22–25]. In the context of modulation recognition, CNN excel at capturing local frequency features through convolutional layers, followed by pooling layers that reduce feature space dimensionality. This enhances their robustness in low Signal-to-Noise Ratio (SNR) environments, making CNN particularly suitable for noisy communication channels. In such conditions, CNN demonstrate high classification performance even when signals are significantly degraded.

Numerous studies demonstrate that CNN hold substantial advantages in modulation recognition tasks. For instance, CNN model has achieved high recognition accuracy under low SNR conditions, showing improved stability in high-noise environments [26–28]. Furthermore, incorporating deep residual networks (ResNet) has enhanced model performance in complex channels, addressing challenges associated with low SNR environments [29,30].

Lightweight CNN structures have been developed to maintain recognition accuracy while reducing computational complexity, making them particularly suitable for embedded systems [31]. Additionally, studies utilizing spectrograms generated from signal data have improved CNN' ability to recognize frequency and phase information, especially in low-SNR classification scenarios [32]. In summary, CNN' feature extraction capabilities and efficient handling of signal matrices make them highly effective in recognizing complex modulated signals.

## Section B: Multi-scale convolutional neural network in modulation recognition

The core idea behind multi-scale algorithms is to process data across various receptive fields by using convolution kernels of different sizes, enabling the model to extract feature information at multiple levels. In signal modulation recognition tasks, signals may display diverse characteristics across different temporal or frequency scales. Multi-scale convolutional neural networks are particularly effective at capturing these variations, significantly enhancing classification accuracy and model robustness.

Multi-Scale CNNs utilize convolution filters of varying sizes to extract hierarchical features from input data. By combining these features with dimensionality reduction through pooling

layers, this approach generates feature vectors that more effectively represent signal characteristics. This multi-scale method is especially beneficial in environments influenced by noise or multipath effects, as it improves the model's ability to generalize under complex channel conditions.

In the field of modulation recognition, multi-scale techniques have been extensively studied and applied, with various methods drawing from each other to demonstrate outstanding performance in different environments. The MSNet-SF model [33] effectively combines traditional statistical features with deep learning, utilizing multi-scale convolutional kernels and sparse connections to enhance accuracy while reducing computational complexity. This approach is particularly effective in distinguishing similar modulation signals, such as QAM16 and QAM64.

Multi-scale convolutional networks also capture hierarchical features through combinations of different convolutional kernel sizes. Based on this, other studies [34] have proposed a multi-scale CNN classifier, transforming signals into constellation diagrams for training and using skip connections to address vanishing gradients, thus maintaining high precision even in noisy environments. This complements MCNet [35], which leverages asymmetric convolutional kernels and residual connections, resulting in both high classification accuracy and strong stability.

Further research [36] adopts a deep CNN architecture with multi-hop connections, extending the receptive field and incorporating multi-hop links to enhance performance across signals of various lengths. Meanwhile, the LSMFF-AMC model [37] combines multimodal feature fusion with long-short range attention, achieving exceptional classification accuracy in complex 6G environments. The LSRA integration allows the model to maintain high recognition accuracy under dynamic channel conditions.

Overall, these multi-scale approaches optimize feature extraction, noise robustness, and computational efficiency from different perspectives, showing complementary structural relationships and collectively enhancing generalization capability in complex signal scenarios.

## Section C: Gated recurrent unit (GRU) in modulation recognition

GRU is an improved variant of the Recurrent Neural Network designed to address issues such as vanishing and exploding gradients that often arise during RNN training. GRUs incorporate update and reset gates, enabling efficient information flow and better handling of sequential data, which is crucial for tasks like signal modulation recognition. GRU's ability to capture temporal dependencies in signals makes it an ideal choice for improving accuracy in modulation classification.

Compared to LSTM networks, GRU has a simpler structure with fewer parameters, resulting in higher computational efficiency, especially when processing long sequences. The following studies have demonstrated GRU's application in modulation recognition and other signal processing tasks: These studies highlight various approaches to utilizing GRU for modulation recognition, emphasizing its advantages in handling temporal data and extracting signal features. The connections among these works primarily reflect:

Structural Enhancements for Optimized Performance: Several studies have focused on structural improvements to GRU for more effective modulation recognition. For instance, Guo et al. [38] proposes an AMR scheme combining predictive correction with dual GRUs to reduce channel errors and computational complexity, while Liu et al. [41] compares LSTM and GRU, integrating high-order cumulants and instantaneous features to boost recognition

accuracy under low SNR. Both studies illustrate efforts to optimize GRU structure, achieving more reliable modulation recognition.

Hybrid Networks for Enhanced Feature Extraction: Studies such as [39] and [42] show the benefits of integrating GRU with other neural networks to enhance modulation recognition accuracy. Njoku et al. [39] combines GRU with shallow CNN and DNN layers to improve feature extraction in cognitive radio services, while Liu et al. [42] implements a dual-stream CNN-GRU architecture with an attention mechanism to classify multiple modulation types accurately. These studies demonstrate how combining GRU with other networks strengthens its capacity to manage complex features.

Feature Fusion Techniques: Various works leverage feature fusion to improve GRU's recognition performance in noisy and complex environments. Li et al. [40] employs ResNeXt and GRU to extract semantic and temporal features, respectively, and uses discriminant correlation analysis for feature fusion, achieving superior performance in complex signal environments. Feature fusion enables models to capture multi-level signal characteristics, which is crucial for robust classification in complex channels.

Adaptation to Complex Channels and Low-SNR Conditions: GRU has shown resilience in handling low-SNR conditions, as highlighted in studies such as [39], [41], and [42]. The unique architecture of GRU makes it adaptable to noise and complex channel environments. When combined with CNN, GRU not only captures temporal dependencies but also spatial features, providing comprehensive support for modulation recognition in challenging conditions. The application of GRU in modulation recognition is underscored by its capacity to capture temporal features while also benefiting from hybrid architectures, feature fusion, and optimized structural enhancements. Together, these methods enable improved robustness and adaptability in complex communication environments, solidifying GRU's role as a valuable component for signal modulation recognition tasks.

## Dataset description

In this study, we utilize three major shortwave communication modulation schemes as the primary data sources for experimentation: CLOVER-2000, 2GALE, and 3GALE. These modulation schemes are widely employed in shortwave communications, particularly in military, emergency communications, and long-distance data transmission. They are known for their robust anti-interference capabilities and resilience in complex electromagnetic environments. Below, we provide a detailed description of each modulation scheme and their internal sub-modulation methods.

### Section A: CLOVER-2000 modulation scheme

CLOVER-2000 is a shortwave communication protocol utilizing multi-carrier modulation, where multiple subcarriers transmit data in parallel, enhancing signal robustness and interference resistance. It is ideal for complex channel conditions in spectrum-limited environments. Typical subcarrier frequencies in CLOVER-2000 are modulated relative to the signal's center frequency.CLOVER-2000 is a shortwave communication protocol utilizing multi-carrier modulation, where multiple subcarriers transmit data in parallel, enhancing signal robustness and interference resistance. It is ideal for complex channel conditions in spectrum-limited environments. Typical subcarrier frequencies in CLOVER-2000 are [625 Hz, 875 Hz, 1125 Hz, 1375 Hz, 1625 Hz, 1875 Hz, 2125 Hz, 2375 Hz], modulated relative to the signal's centre frequency.

1. BPSM (Binary Phase Shift Modulation): Input symbols are mapped as follows:

$$s[n] = (-1)^{x[n]} \tag{1}$$

where $x[n] \in \{0, 1\}$, representing phase shifts of $0^o$ and $180^o$.

2. QPSM (Quadrature Phase Shift Modulation): Input symbols $x[n] \in \{0, 1, 2, 3\}$ are mapped to the complex plane as:

$$s[n] = e^{j\frac{\pi}{2}x[n]} \tag{2}$$

3. 8PSM (8-Phase Shift Modulation): Input symbols $x[n] \in \{0, 1, ..., 7\}$ are mapped as:

$$s[n] = e^{j\frac{\pi}{4}x[n]} \tag{3}$$

allowing for 8 different phase shifts.

4. 8P2A and 16P4A: These higher-order modulations combine phase and amplitude shifts, represented as:

$$s[n] = A_n e^{j\theta_n} \tag{4}$$

where $A_n$ represents amplitude and $\theta_n$ the phase change.

Subcarrier frequencies are determined as:

$$f_k = f_c + k\Delta f \tag{5}$$

where $f_c$ is the center frequency, $\Delta f$ is the frequency spacing, and $k$ is the subcarrier index. The final modulated signal is the sum of all subcarrier signals:

$$s(t) = \sum_k s_k(t) \cos(2\pi f_k t + \phi_k) \tag{6}$$

where $s_k(t)$ represents the symbol on subcarrier $k$, $f_k$ is the subcarrier frequency, and $\phi_k$ is the carrier phase.

This multi-carrier scheme significantly enhances CLOVER-2000's resistance to noise, fading, and interference. Fig 1 illustrates the time-domain and frequency-domain characteristics of CLOVER-2000's BPSM signals.

## Section B: 2GALE modulation scheme

The 2GALE modulation scheme, specifically designed for shortwave communication, utilizes a 3-bit symbol set, with each symbol mapped onto an 8-point constellation diagram, corresponding to frequencies between 750 Hz and 2500 Hz. This frequency mapping allows efficient data transmission within limited bandwidth.

The frequency mapping for each bit symbol is given as

$$f_k = 750 + k \cdot 250 \, \text{Hz}, \quad k = 0, 1, ..., 7 \tag{7}$$

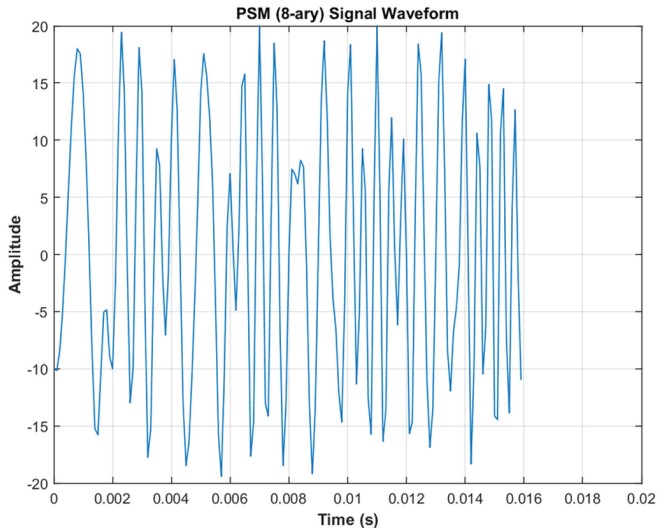
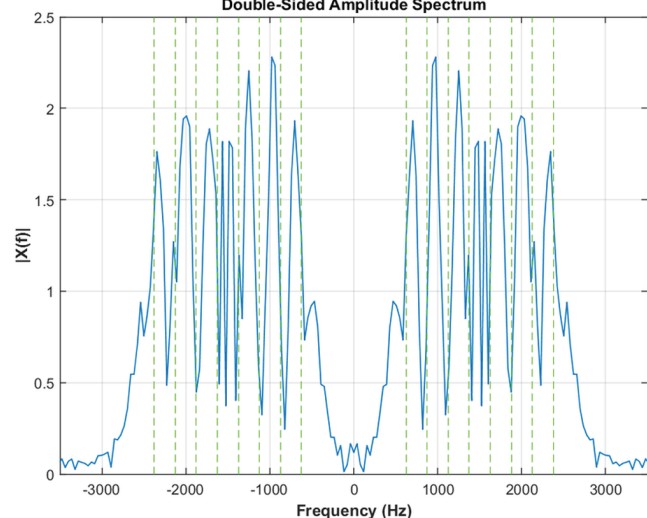

**Fig 1. CLOVER-2000-BPSM's time-domain and frequency-domain.**

where $k$ represents the bit symbol index. The modulated signal is represented as

$$s(t) = \sum_{n=0}^{N-1} \cos(2\pi f_s nt) \tag{8}$$

Here, $f_k$ is the modulated carrier frequency. This modulation scheme is robust against interference, making it suitable for long-distance transmission over shortwave channels. Fig 2 shows the time-domain, frequency-domain, and time-frequency characteristics of the 2GALE signal.

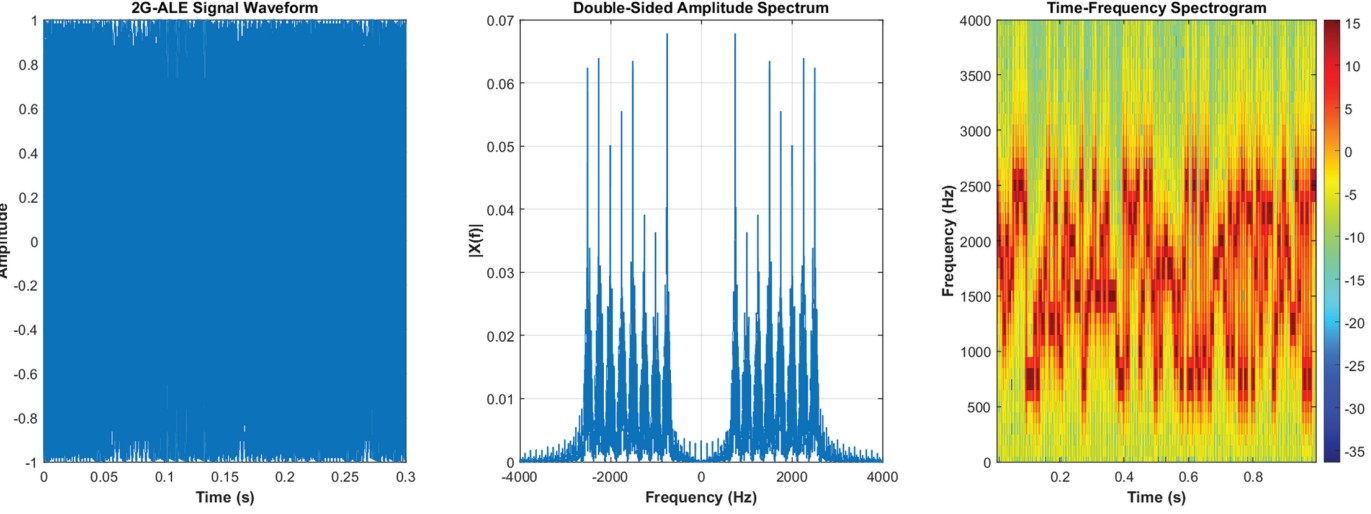

**Fig 2. 2GALE's time-domain and frequency-domain and time-frequency characteristics.**

## Section C: 3GALE modulation scheme

The 3GALE modulation scheme is part of the third-generation shortwave communication system, primarily used for tasks involving link establishment and data transmission. The system defines six burst waveforms (BW), each tailored to specific communication requirements. These waveforms employ 8-phase shift keying modulation (8-PSK) and are transmitted via a 1800 Hz carrier tone, achieving a data rate of 2400 characters per second.

Table 1 provides details on the characteristics of the six types of burst waveforms, covering various aspects such as usage, burst duration, data payload, preamble, encoding and interleaving methods, data format, and effective coding rate. These parameters outline the specific roles and configurations of each waveform, facilitating efficient data transmission and resilience in different communication scenarios.

Based on the characteristics of the six burst waveforms listed in Table 1, corresponding 3GALE signals can be generated by following each waveform's specific details and outlined steps.

*Payload generation*: Based on the payload size specified for each waveform, generate a random binary sequence of corresponding length to serve as the payload. For each waveform B, generate a random binary payload sequence.

$$d = [d_0, d_1, \ldots, d_n] \tag{9}$$

where *n* corresponds to the specific bit length required for each waveform as specified in Table 1.

**Table 1. Characteristics of burst waveforms.**

| Waveform | Application | Burst Duration | Payload | Preamble | FEC Coding | Interleaving | Effective Coding Rate |
|---|---|---|---|---|---|---|---|
| BW0 | Robust LSU PDU | 613.33 ms, 1472 PSK symbols | 26 bits | 160ms, 384 PSK symbols | Convolutional Code, Rate 1/2, K=7 (No flush bits) | 4×13 groups | 1/96 |
| BW1 | Service Management PDU / HDL ACK PDU | 1.30667 s, 3136 PSK symbols | 48 bits | 240ms, 576 PSK symbols | Convolutional Code, Rate 1/3, K=9 (No flush bits) | 16×9 groups | 1/144 |
| BW2 | HDL Service Data PDU | 126.67 + (n×400) ms, 304 + (n×960) PSK symbols, n=3,6,12,24 | n×1881 bits | 26.67ms, 64 PSK symbols | Convolutional Code, Rate 1/4, K=8 (7 flush bits) | 32 unknown / 16 known symbols | Variable: 1/1, 1/2, 1/3, 1/4 |
| BW3 | LDL Service Data PDU | 373.33 + (n×13.33) ms | 8n+25bit | 266.7 ms, 640 PSK symbols | Convolutional Code, Rate 1/2, K=7 (7 flush bits) | Convolutional grouping | Variable: 1/12, 1/24 |
| BW4 | LDL Acknowledgement PDU | 640.00 ms | 2 bits | - | - | Quaternary Walsh function | 1/1920 |
| BW5 | Fast LSU PDU | 1.01333 s, 2432 PSK symbols | 50 bits | 240.00ms | Convolutional Code, Rate 1/2, K=7 (No flush bits) | 10×10 groups | 1/96 |

Table notes: Definitions of all waveform types are detailed in the text.

*Convolutional encoding*: The binary *d* is convolutionally encoded to introduce redundancy for error correction. Using the convolutional encoding formula:

$$c_i = \sum_{j=0}^{K-1} g_j \cdot d_{i-j} \quad (\text{mod } 2) \tag{10}$$

where $g_i$ represents the generator polynomial coefficients, and *K* is the constraint length specific to the waveform's error-correction scheme. This step generates an encoded sequence $c = [c_0, c_1, ..., c_m]$, which has greater length than the original *d* due to the convolutional redundancy.

*Symbol mapping*: Using modulation methods such as PSK, QAM, etc., the encoded binary sequence *c* is mapped to complex-valued symbols $s_i$:

$$s_i = e^{j\frac{2\pi k_i}{M}} \tag{11}$$

where *M* is the modulation order (e.g., 8 for 8-PSK), $k_i$ is the integer representation of bits mapped from $c_i$. The sequence $s = [s_0, s_1, ..., s_m]$ represents the modulated symbols in the complex plane.

*Scrambling processing*: To randomize and reduce long repetitive sequences, each symbol in s is scrambled with a scrambling sequence $seq_{scrambling}$ using a bitwise XOR operation:

$$d_{scrambled} = d \oplus seq_{scrambling} \tag{12}$$

where $d_{scrambled}$ is the scrambled payload, now ready for modulation.

*Signal modulation*: Each scrambled symbol $s_i$ undergoes carrier modulation to generate the final transmitted signal *x(t)*:

$$x(t) = \mathbb{R}\left\{ s_i \cdot e^{j2\pi f_c t} \right\} \tag{13}$$

where $f_c$ is the carrier frequency, and $\mathbb{R}$ denotes taking the real part of the complex modulated signal.

Using the equations derived and the characteristics listed in Table 1, the signals for burst waveforms BW0 through BW5 can be generated. These generated waveforms, shown in Fig 3–Fig 8, illustrate the simulation results that replicate the distinct signal structures and modulation characteristics defined for each waveform type.

Fig 3 depicts BW0's time-domain, frequency-domain, and time-frequency characteristics, while Fig 4 highlights BW1's time-domain, frequency-domain, and time-frequency characteristics. Fig 5 shows BW2's time-domain, frequency-domain, and time-frequency characteristics, and Fig 6 presents BW3's time-domain, frequency-domain, and time-frequency characteristics. Fig 7 displays BW4's time-domain, frequency-domain, and time-frequency characteristics, finnaly, Fig 8 displays BW5's time-domain, frequency-domain, and time-frequency characteristics, providing a comprehensive representation of the 3GALE burst signals under various configurations. Each figure also emphasizes the modulation, scrambling, and coding properties specified for the corresponding waveform.

## Section D: USRP

To evaluate the MSC-GRU model under realistic shortwave channel conditions, we employed the USRP B210 (Universal Software Radio Peripheral) for shortwave signal transmission and

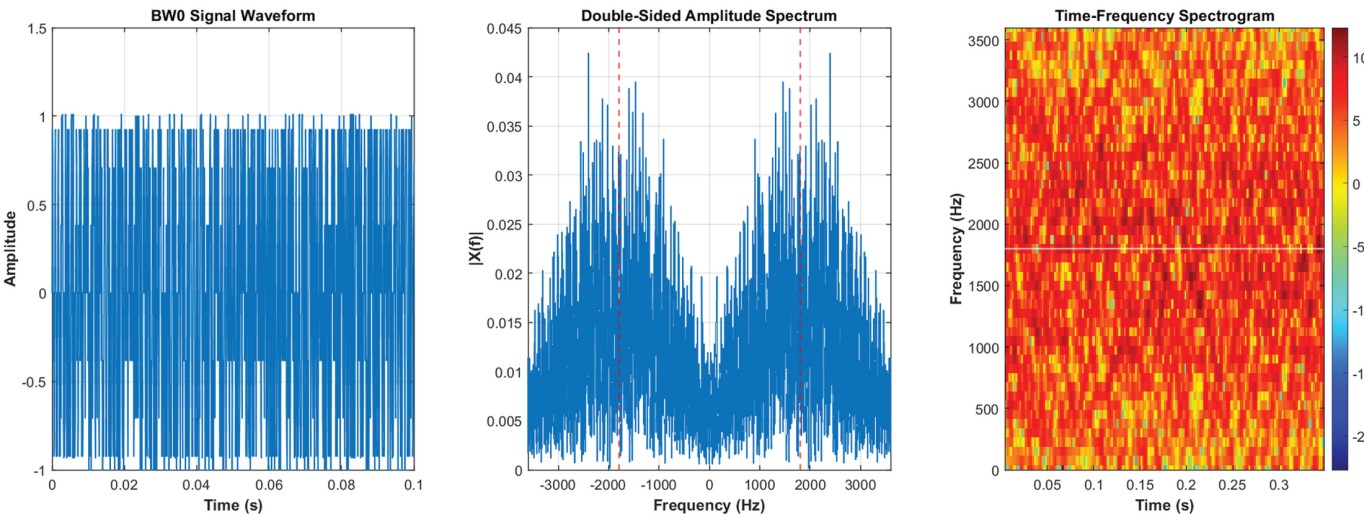

**Fig 3. BW0's time-domain and frequency-domain and time-frequency characteristics.**

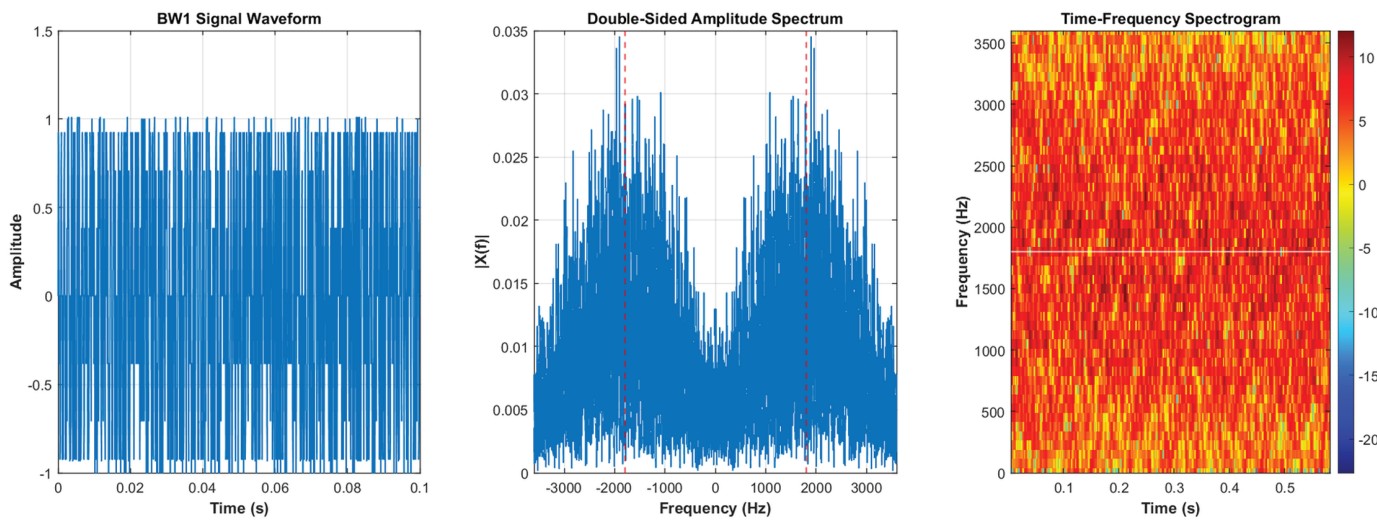

**Fig 4. BW1's time-domain and frequency-domain and time-frequency characteristics.**

reception. This hardware platform supports real-time signal acquisition, making it suitable for capturing signal variations under different environmental conditions.

The USRP system was configured to transmit shortwave signals modulated according to the CLOVER-2000, 2GALE, and 3GALE protocols. Signals were generated using GNU Radio and transmitted over-the-air in both indoor and outdoor environments. In the indoor setting, multipath fading, wall reflection, and electromagnetic interference were prevalent. In contrast, the outdoor environment was characterized primarily by free-space path loss and natural noise sources.

The USRP transmitter and receiver were placed at distances ranging from 0 meters to 7 meters. The transmission frequency was set in the 3–5 MHz range to simulate typical

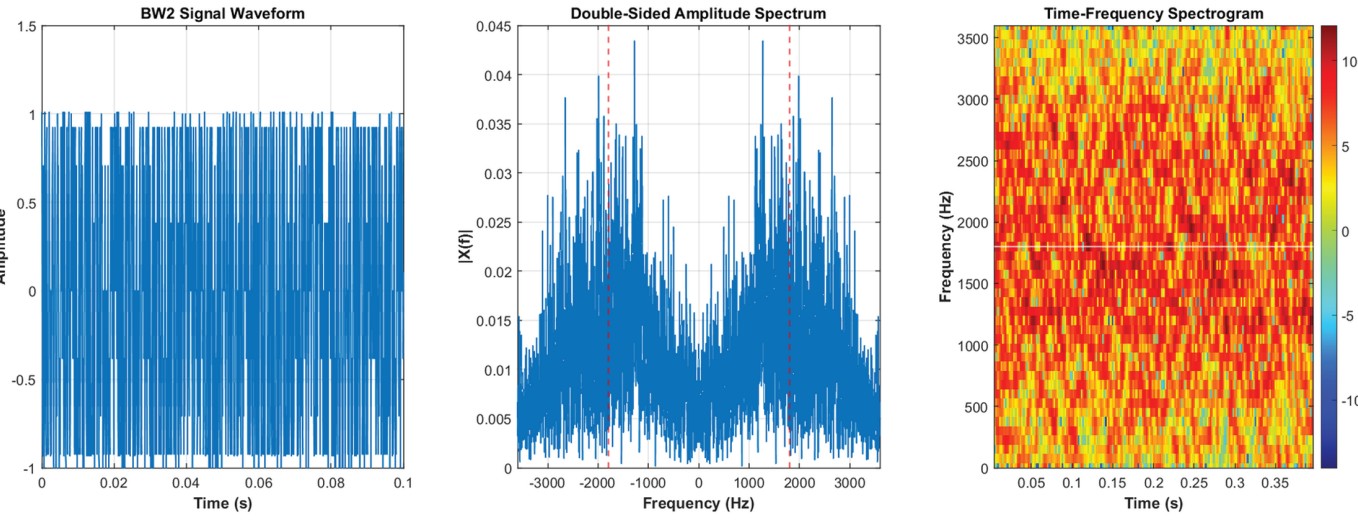

**Fig 5. BW2's time-domain and frequency-domain and time-frequency characteristics.**

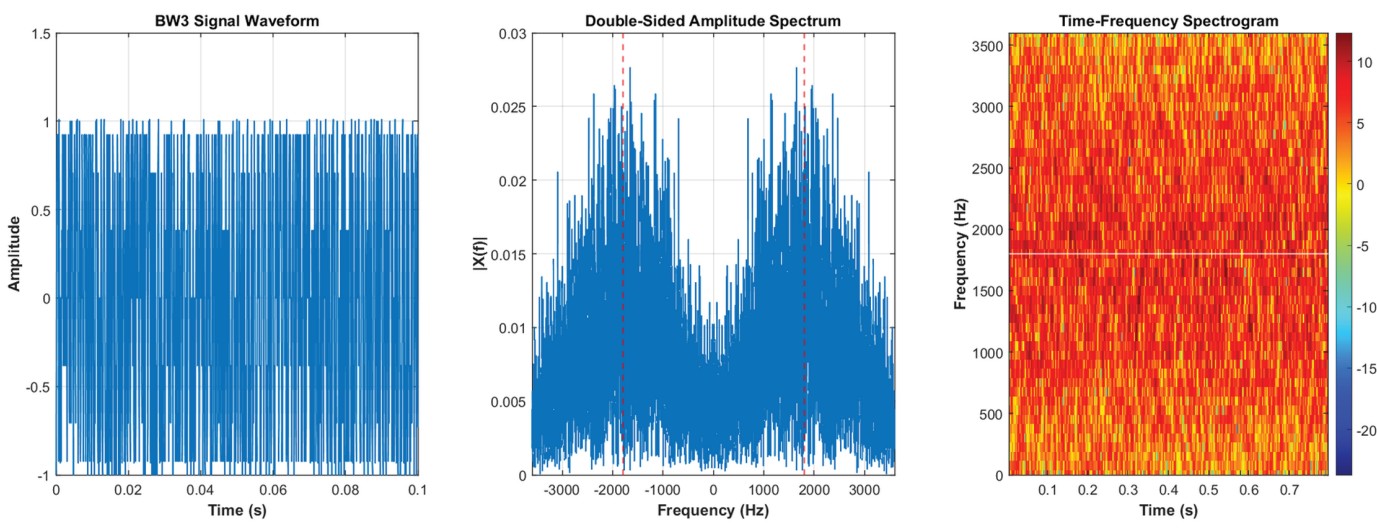

**Fig 6. BW3's time-domain and frequency-domain and time-frequency characteristics.**

shortwave conditions. The sampling rate was 5 MHz, and adaptive gain control was enabled to ensure signal stability. Each received signal was truncated to a length of 1000 points, consistent with the simulated dataset format.

These real-world signal samples were processed using the same MSC-GRU architecture, and the results are reported in Section C of Results. This practical dataset enables us to evaluate the model's performance under channel impairments such as fading, interference, and time-varying SNR, thereby confirming the robustness and applicability of the proposed model in realistic shortwave scenarios.

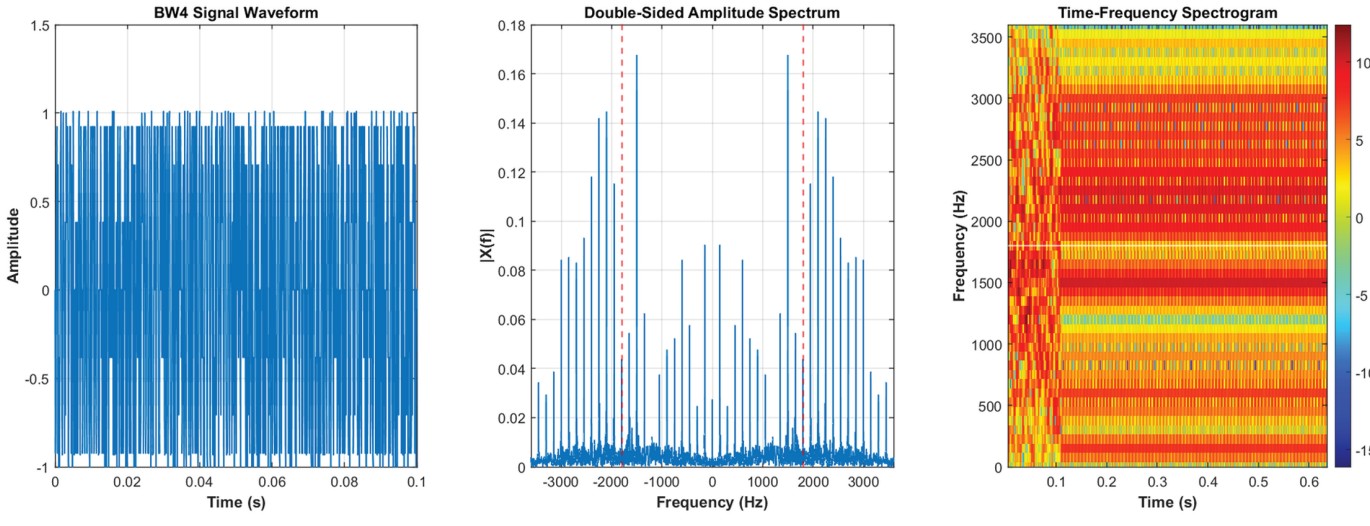

**Fig 7. BW4's time-domain and frequency-domain and time-frequency characteristics.**

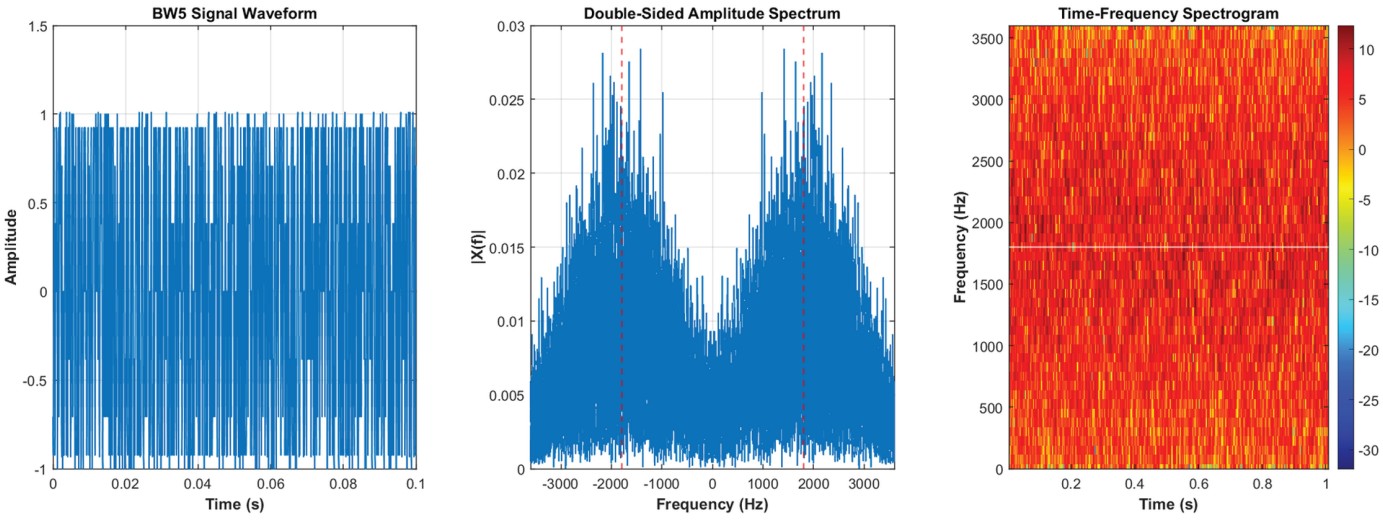

**Fig 8. BW5's time-domain and frequency-domain and time-frequency characteristics.**

## Method

The overall architecture of the model is designed based on a deep understanding of shortwave signal characteristics. It integrates Spatial Feature Extraction model, Multi-Scale Convolutional model, and Temporal Feature Extraction model, as illustrated in Fig 9.

*Spatial feature extraction*: The spatial feature extraction module consists of a series of parallel feature extractors, denoted as $P_1$, $P_2$, and $P_3$, each responsible for capturing spatial dependencies within the input signal. These extractors are implemented as convolutional layers,

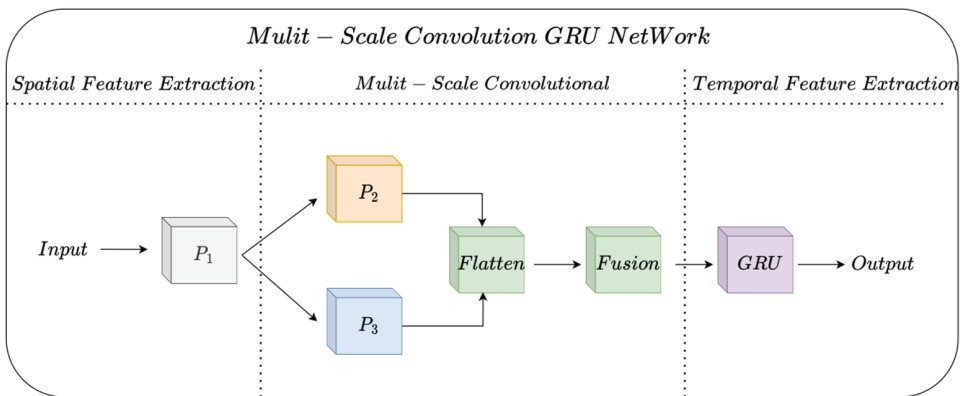

**Fig 9. Mulit-scale convolution GRU netWork structure.**

where the output $P_i$ of each extractor can be defined as:

$$P_i = f(W_i * X + b_i) \tag{14}$$

where $*$ denotes the convolution operation, $W_i$ represents the filter weights, $X$ is the input signal, $b_i$ is the bias term, and $f$ is an activation function, typically ReLU. By processing the input signal through multiple spatial filters, this module enhances the model's ability to capture localized spatial features that are critical for identifying signal patterns.

*Multi-scale convolutional module*: To capture signal features at various scales, a multi-scale convolutional module is employed. This module processes the output of the spatial feature extractors with multiple convolutional layers of varying kernel sizes. The output M of the multi-scale convolutional layer is computed as follows:

$$M = \sum_{k=1}^{K} f(W_k * P + b_k) \tag{15}$$

where $K$ represents the number of scales, each associated with a unique filter $W_k$ and bias term $b_k$, applied to the combined output $P$ from the spatial feature extractors. This approach enhances the model's ability to capture features at different levels of granularity, which is essential for handling the variability in shortwave signal characteristics.

*Temporal feature extraction*: The GRU-based temporal feature extraction module is utilized to capture sequential dependencies in the signal. The GRU processes the fused feature vector $F$ over time steps, generating a sequence of hidden states $h_t$ that encapsulate temporal information. The update function for each GRU cell can be represented as:

$$h_t = GRU(h_{t-1}, F) \tag{16}$$

where $h_t$ is the hidden state at time $t$, retaining information about previous inputs and effectively modelling temporal dependencies within the sequence. This is particularly important for signals with sequential patterns, such as those in shortwave communication.

Combining spatial feature extraction, multi-scale convolution, and temporal sequence modeling, the proposed architecture is designed to comprehensively capture both local and

sequential characteristics of shortwave signals. The resulting model offers robust performance in shortwave signal classification by effectively leveraging both spatial and temporal dependencies within the data.

### Section A: Spatial feature extraction model

The Residual Network serves as a critical sub-module within the Multi-Scale Convolutional Network, responsible for enhancing the spatial feature extraction process by allowing deeper feature learning without encountering the vanishing gradient problem. The structure, as shown in Fig 10, consists of stacked convolutional layers with residual connections, enabling the network to learn more complex features across multiple scales.

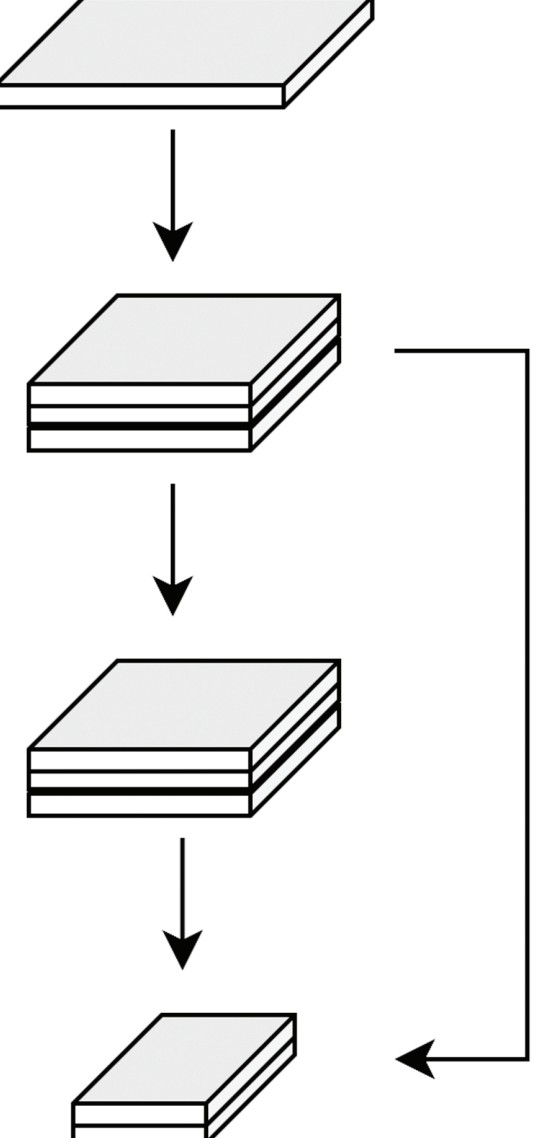

**Fig 10. Residual structure.**

In this residual block, the input $x_{input}$ passes through two convolutional layers with weights $W_1$ and $W_2$ and biases $b_1$ and $b_2$. Each convolutional layer applies an activation function $f$, often ReLU, before generating the final output $x_{residual}$ of the block:

$$x_{residual} = f(x_{input} * W_1 + b_1) * W_2 + b_2 + x_{input} \tag{17}$$

This formula encapsulates the operation of the residual network by showing that the final output $x_{residual}$ is a combination of the transformed features and the original input. The residual connection effectively creates a shortcut for the gradient flow, allowing deeper layers to learn more intricate spatial features without degradation in performance.

## Section B: Multi-scale convolutional

The Multi-Scale Convolutional Network (MSCNet) in this model uses two types of convolutional kernels—$2 \times 2$ and $4 \times 4$—to capture spatial features at different resolutions, as shown in Fig 11. By combining these kernels, the network achieves a multi-scale feature representation that enhances spatial analysis across diverse receptive fields. This approach is essential for capturing both localized and broader contextual information in shortwave signals.

The $2 \times 2$ kernel focuses on finer, localized details, making it suitable for detecting subtle variations and intricate patterns within the signal. Conversely, the $4 \times 4$ kernel captures more extensive regions within the input feature map, allowing for a broader, less detailed view that is beneficial for understanding overall structural features. These two scales effectively complement each other:

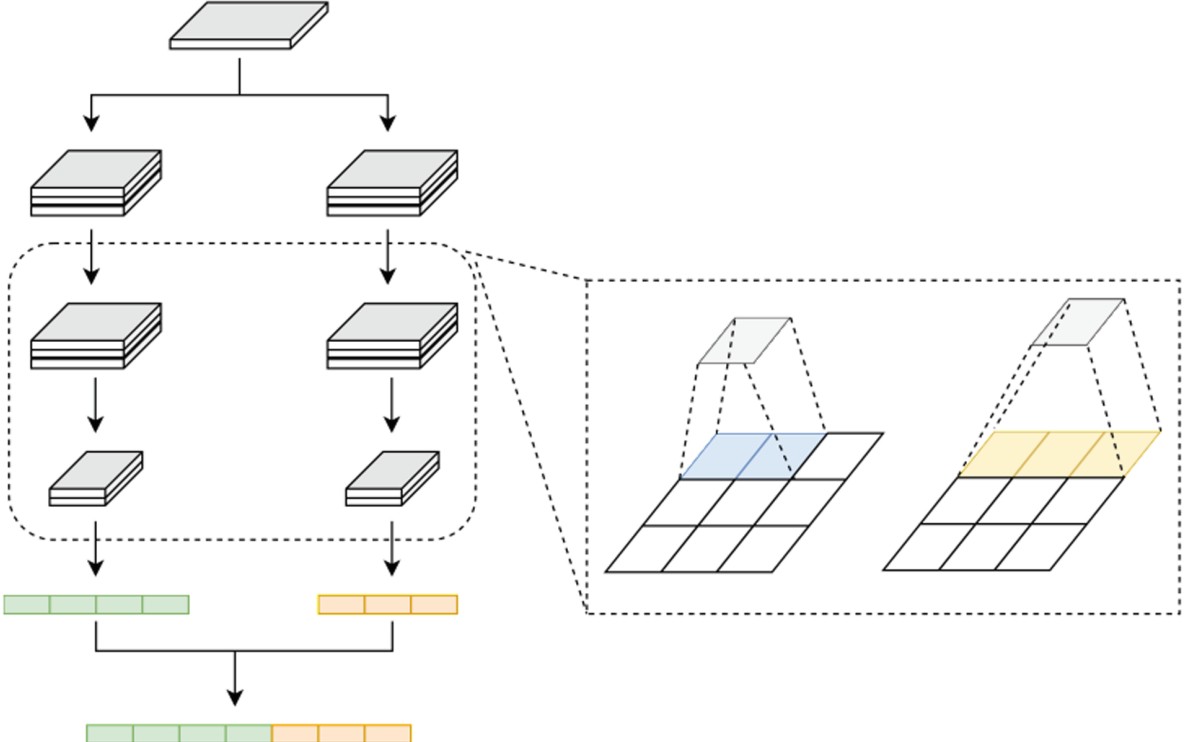

**Fig 11. Multi-scale network structure.**

2 × 2 Convolutional kernel: Offers a small receptive field that captures high-resolution, detailed features.

4×4 Convolutional kernel: Provides a larger receptive field, capturing more global spatial characteristics.

After applying the 2×2 and 4×4 convolutions on the input feature maps, the resulting feature maps undergo a flattening process to convert each feature map into a one-dimensional representation. This operation is performed independently on the feature maps from each kernel size, transforming them from spatial feature maps into vectors.

Following flattening, the vectors obtained from the 2×2 and 4×4 convolution operations are concatenated along the depth dimension. This fusion strategy enhances the model's multi-scale feature representation by combining fine-grained details from the 2×2 convolution with broader spatial information from the 4×4 convolution, yielding a unified feature vector for subsequent layers.

If $F_{2×2}$ and $F_{4×4}$ represent the feature maps obtained from the 2×2 and 4×4 convolutions, respectively, then the flattening and concatenation process can be mathematically represented as:

$$F_{fusion} = Concat(Flatten(F_{2×2}), Flatten(F_{4×4})) \tag{18}$$

where $F_{fusion}$ is the combined one-dimensional feature vector that integrates the multi-scale information. This fused representation is then used as input for the subsequent network layers, such as the temporal feature extraction module, enhancing the model's capability to leverage both local and global spatial characteristics.

In this study, the generated signals were truncated to a length of 1000 to simulate real-world conditions where it is not possible to capture the entire signal. The truncated length was reshaped into a 20×50 matrix, converting the sequence into an image for signal analysis from an image-based perspective. The input format is shown in Fig 12.

The image shows the distribution of signal values in a 20×50 matrix format. The color bar provides the range of signal values, transitioning from purple (negative values) to yellow (positive values), indicating the variation of signal values across different points. Visually, the striped structure in the image suggests that the signal may have some periodic or regular oscillations. These stripes are usually a result of frequency components, and when the sequential data is rearranged into an image, it retains certain time-based characteristics. This provides foundational information for subsequent algorithm processing.

Fig 13 shows the output feature maps from the first multi-scale convolution layer. The image is arranged in a 3×6 grid, displaying 16 channel outputs in total (although the actual number of channels in the algorithm may not be strictly equal to 16). Each small image represents features extracted in a different channel after convolution, illustrating the varied response patterns of the convolution layer to the input signal.

These feature maps demonstrate the convolutional layer's response to the signal in different channels, indicated by the color variation (ranging from purple to yellow), which represents different signal intensity regions. Some of the small images show noticeable stripes or textures, likely related to the periodicity or frequency components of the signal. Through these feature maps, it is evident that the convolutional layer selectively extracts different signal patterns, aiding the model in more effectively identifying distinct features in the signal at subsequent layers.

Fig 14 shows the output feature maps of the second multi-scale convolution layer. Compared to the first multi-scale convolution layer, the second layer uses a larger convolution

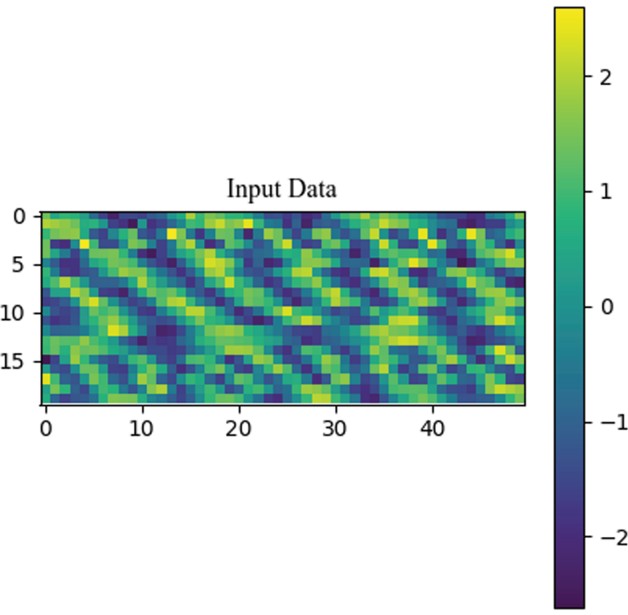

**Fig 12. Transform the sequence into image information.**

# Mulit-Scale Convolution 1 outputs

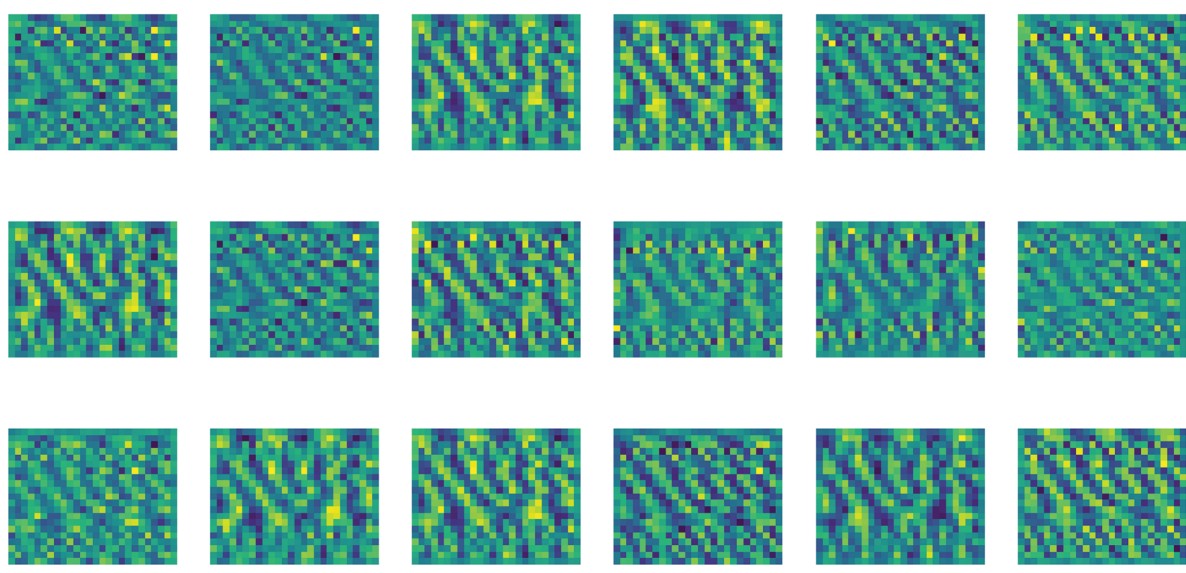

**Fig 13. Output feature maps from the first multi-scale convolution.**

## Mulit-Scale Convolution 2 outputs

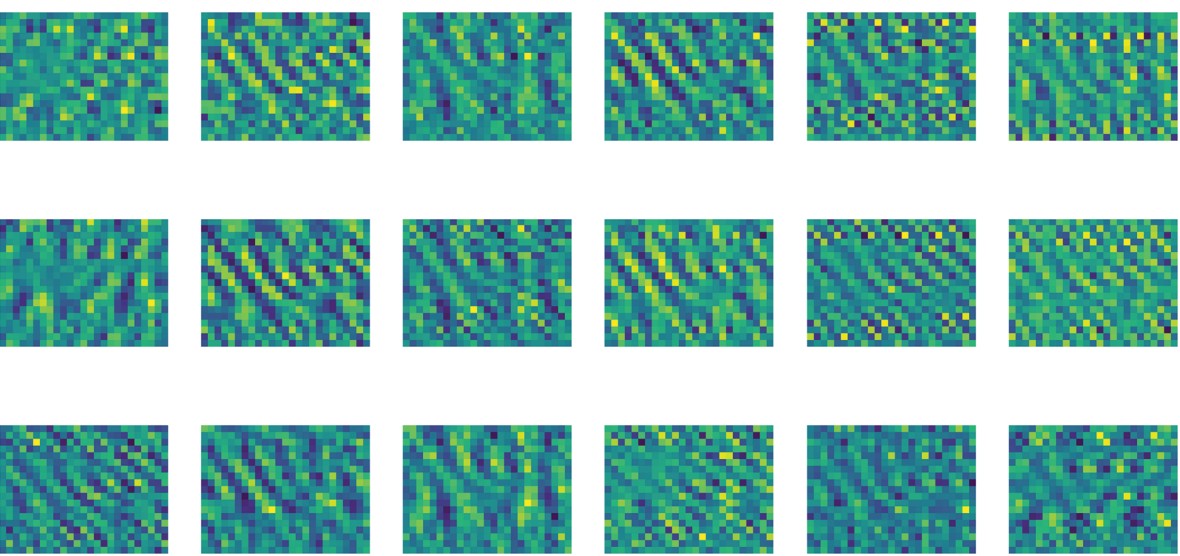

**Fig 14. Output feature maps from the second multi-scale convolution.**

kernel, allowing it to capture information within a larger receptive field and thereby extract broader signal features. By employing a larger kernel, this second multi-scale convolution layer can integrate patterns and features across a wider area, enhancing its ability to capture more global characteristics of the signal.

This layer exhibits more refined structures, capturing higher-level features within the signal. The larger kernel makes the layer more sensitive to the overall structure of the signal, enabling it to focus on cross-regional correlations. The bright and dark areas in these feature maps are more clearly defined, indicating that this layer's convolution operation improves the network's focus and resolution capabilities, providing a more precise foundation for subsequent network layers.

### Section C: Temporal feature extraction

The GRU is utilized in this model as the temporal feature extraction layer, as shown in Fig 15, primarily designed to capture sequential dependencies within the fused multi-scale feature maps. As a type of recurrent neural network, the GRU is particularly effective at handling sequential data, such as time-series information, by maintaining a balance between capturing long-term dependencies and reducing the risk of vanishing gradients. This makes it ideal for tasks involving shortwave signal sequences.

The GRU contains two main gates: the update gate and the reset gate. These gates control the flow of information, allowing the model to retain relevant past information while discarding irrelevant details. The following equations represent the internal computations within the GRU:

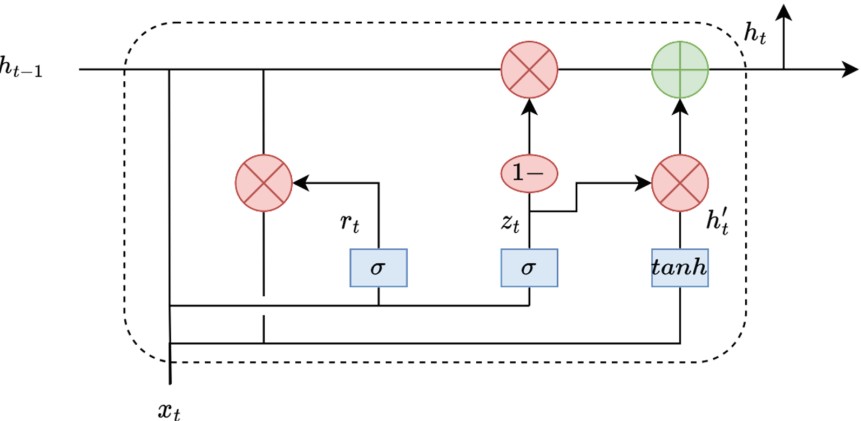

**Fig 15. GRU network structure.**

The update gate $Z_t$ determines how much of the previous hidden state $h_{t-1}$ should carry over to the current state. It is computed as:

$$z_t = \sigma\left(W_Z \cdot [x_t, h_{t-1}] + b_Z\right) \tag{19}$$

where $W_Z$ and $b_Z$ are the weights and biases, $x_t$ is the input, and $\sigma$ is the sigmoid activation function.

The reset gate $r_t$ decides how much of the past information to forget. It is calculated by:

$$r_t = \sigma\left(W_r \cdot [x_t, h_{t-1}] + b_r\right) \tag{20}$$

The candidate hidden state $\tilde{h}_t$ combines the new input with the past hidden state, modulated by the reset gate:

$$\tilde{h}_t = \tanh\left(W \cdot [x_t, (r_t \odot h_{t-1})] + b\right) \tag{21}$$

The final hidden state $h_t$ is an interpolation between the previous hidden state $h_{t-1}$ and the candidate hidden state $\tilde{h}_t$, controlled by the update gate:

$$h_t = (1 - z_t) \odot h_{t-1} + z_t \odot \tilde{h}_t \tag{22}$$

where $\sigma$ is the sigmoid activation function, $\odot$ denotes element-wise multiplication. tanh is the hyperbolic tangent activation function, which helps in regulating the values of $\tilde{h}_t$ between -1 and 1.

In this model, the GRU layer follows the flattening and fusion of multi-scale feature maps, as shown in Fig 10. The GRU's role is to aggregate temporal information across these fused spatial features, thus enabling the model to learn sequential patterns within the signal data, which is crucial for applications in shortwave communication signals that exhibit time dependencies.

Fig 16 shows the output after the GRU layer. The line plot depicts the changes in GRU output values across different time steps. From the plot, it can be observed that the GRU output varies significantly at different time steps, with noticeable oscillations. The rapid alternation

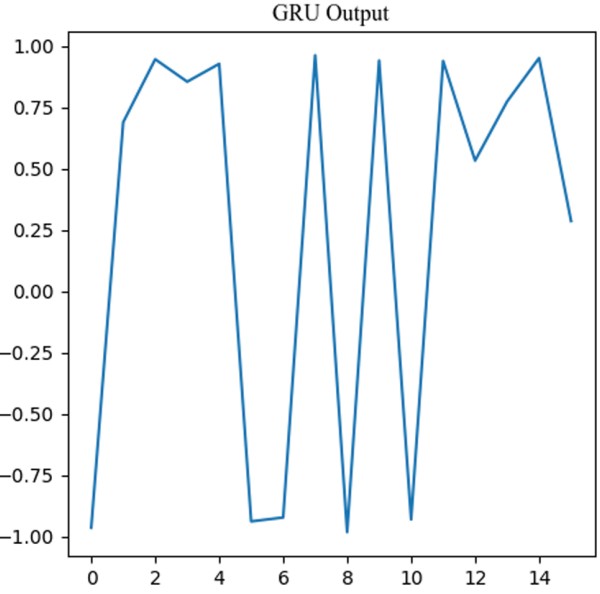

**Fig 16. Output of GUR.**

between positive and negative values suggests that the GRU has learned certain feature patterns within the signal sequence, capturing periodicity and trend variations in the signal. These temporal features will serve as inputs to the subsequent fully connected layer, providing a foundation for the final classification or recognition task. Here, the GRU output offers a summary of features for the following classification layer, representing an aggregation of global temporal information from the signal.

## Result

As described in the methodology section, the dataset used in this study comprises 45,000 labeled signal segments, each with a fixed length of 1000 points. This design simulates real-world scenarios where signals are often partially captured due to transmission interruptions or limited observation windows. The dataset includes a balanced distribution across protocol types and modulation formats, ensuring diversity and representativeness. Although fixed-length truncation may introduce minor accuracy degradation, it ensures consistency in model input, simplifies training convergence, and aligns with practical constraints for real-time deployment.

To validate the effectiveness of the proposed MSC-GRU model, we conduct comparative experiments against two representative baseline models:

1. A standard CNN model, which applies conventional convolutional layers for spatial feature extraction without temporal modeling.
2. A Multi-Scale CNN model, which incorporates convolutional kernels of varying sizes to capture multi-resolution features but lacks temporal sequence handling.

These baselines are commonly used in signal recognition tasks and serve as robust references for performance comparison.

The experiments were executed on a workstation running Windows 11, equipped with an Intel(R) Core(TM) i7-8750H CPU @ 2.20GHz and an NVIDIA GeForce GTX 1050 Ti GPU. The development environment consisted of PyCharm 2021.3.1 as the integrated development platform and Python 3.8 as the programming language.

## Section A: Recognition of three shortwave protocols

In this section, we analyze the performance of the CNN, Multi-Scale CNN, and Multi-Scale CNN-GRU models for identifying three shortwave communication protocols across a range of SNR conditions. Fig 17–Fig 20 depict the recognition accuracy of these models using five-fold cross-validation under SNR levels ranging from –20 dB to 20 dB.

The proposed Multi-Scale Convolutional GRU (MSC-GRU) model consists of two main components: a dual-branch multi-scale convolutional encoder and a bidirectional GRU-based sequence processor. The detailed architecture is outlined in Table 2.

*Multi-scale convolution encoder*: Two convolutional branches are constructed in parallel. Branch 1 uses a $2 \times 2$ kernel and Branch 2 uses a 4x4 kernel. Both branches pass the input through five successive residual blocks. Each residual block consists of a convolutional layer, batch normalization, ReLU activation, and stride-2 downsampling. After the final block, each branch outputs a $1 \times 1$ feature map with 8 channels, which is then flattened.

*Feature fusion and attention*: The flattened outputs from both branches are concatenated to form a 16-dimensional feature vector. This vector passes through a self-attention layer to learn feature weights dynamically.

*GRU temporal modeling*: The attention-enhanced feature is then processed by a bidirectional GRU layer (hidden size = 16), capturing temporal dependencies in sequential data. This is critical for robust performance under conditions of multipath fading and signal overlap.

*Output layer*: A fully connected linear layer maps the GRU output to three protocol classes. Dropout (rate = 0.3) is applied after the GRU layer to prevent overfitting.

All models were trained using categorical cross-entropy loss and the Adam optimizer (learning rate = 0.001, batch size = 128, 100 epochs with early stopping). Hyperparameters were selected via grid search on validation folds during five-fold cross-validation.

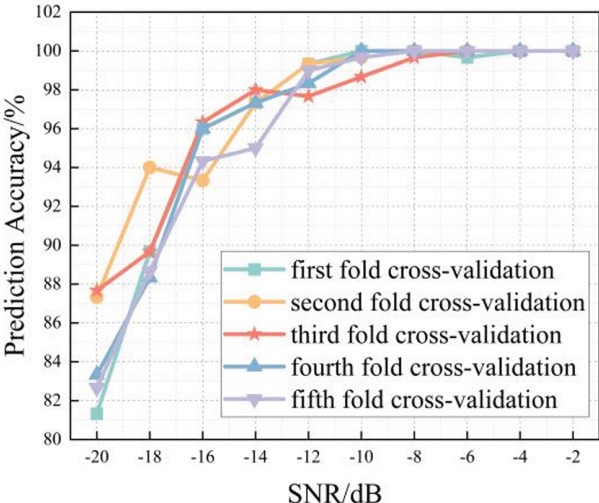

**Fig 17. Five-fold cross-validation results of CNN algorithm at different SNRs.**

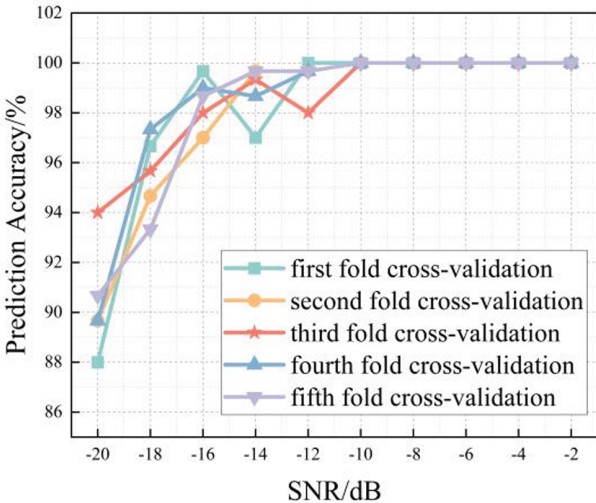

**Fig 18. Five-fold cross-validation results of mulit-scale CNN algorithm at different SNRs.**

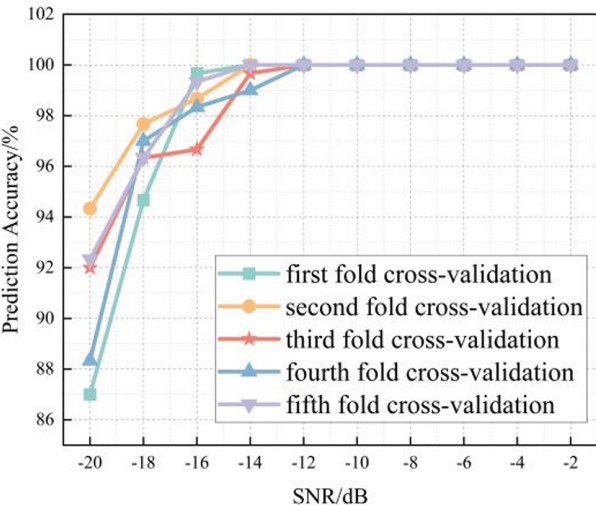

**Fig 19. Five-fold cross-validation results of mulit-scale convolution GRU algorithm at different SNRs.**

Fig 17 presents the accuracy of the standard CNN model across different SNR levels using five-fold cross-validation. The results indicate that the CNN model achieves robust performance for SNR ≥ 0 dB, with accuracy stabilizing close to 100%. However, in low-SNR environments (SNR < 0 dB), the model exhibits considerable variability across folds, with accuracy dropping to approximately 80% at –20 dB. This fluctuation, especially in the –10 dB to –20 dB range, suggests that the basic CNN architecture struggles to consistently extract meaningful features in the presence of strong noise.

Compared to the Multi-Scale CNN and MSC-GRU models discussed later, the standard CNN lacks the capability to effectively capture features at multiple scales or leverage temporal dependencies. This results in a limited adaptability under noisy conditions.

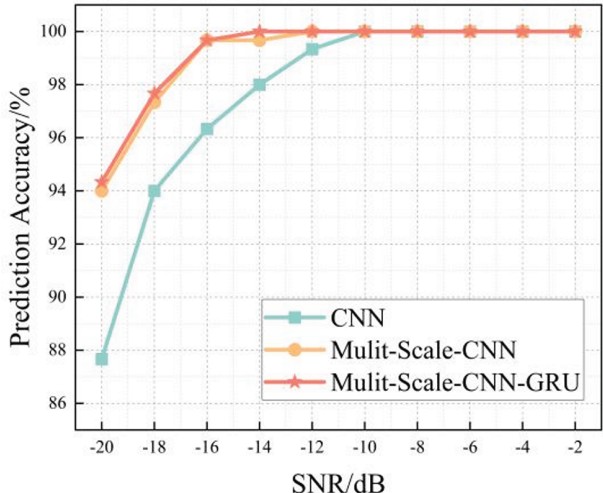

**Fig 20. Comparison of recognition rates for three algorithms at different SNRs.**

**Table 2. Multi-scale convolution GRU model parameters.**

| Layer/Block | Operation/Kernel Size | Stride | Input Shape | Output Shape |
|---|---|---|---|---|
| Input | - | - | (1,1,20,50) | (1,1,20,50) |
| Branch 1 | Conv2d (2 × 2) | (1,2) | (1,1,20,50) | (1,256,21,25) |
| Residual Block 1 | (2 × 2) | 2 | (1,256,21,25) | (1,128,11,13) |
| Residual Block 2 | (2 × 2) | 2 | (1,128,11,13) | (1,64,6,7) |
| Residual Block 3 | (2 × 2) | 2 | (1,64,6,7) | (1,32,3,4) |
| Residual Block 4 | (2 × 2) | 2 | (1,32,3,4) | (1,16,2,2) |
| Residual Block 5 | (2 × 2) | 2 | (1,16,2,2) | (1,8,1,1) |
| Flatten Branch 1 | – | – | (1,8,1,1) | (1,8) |
| Branch 2 | Conv2d (4 × 4) | (1,2) | (1,1,20,50) | (1,256,19,25) |
| Residual Block 1 | (4 × 4) | 2 | (1,256,19,25) | (1,128,10,13) |
| Residual Block 2 | (4 × 4) | 2 | (1,128,10,13) | (1,64,5,7) |
| Residual Block 3 | (4 × 4) | 2 | (1,64,5,7) | (1,32,3,4) |
| Residual Block 4 | (4 × 4) | 2 | (1,32,3,4) | (1,16,2,2) |
| Residual Block 5 | (4 × 4) | 2 | (1,16,2,2) | (1,8,1,1) |
| Flatten Branch 2 | – | – | (1,8,1,1) | (1,8) |
| Concatenate | Concatenate (Flattened Outputs) | - | (1,8)+(1,8) | (1,16) |
| Attention Layer | Self-Attention | – | (1,1,16) | (1,1,16) |
| GRU Layer | GRU | – | (1,1,16) | (1,1,16) |
| Fully Connected Layer | Linear | - | (1,16) | (1,3) |

Table notes: This table describes the detailed parameters of each layer in the Multi-Scale Convolution GRU model.

Furthermore, a threshold performance observation reveals that the CNN model surpasses the 90% accuracy mark starting from SNR = -6 dB, indicating its sensitivity to signal quality. Below this threshold, the model's predictions become less reliable and exhibit instability across different validation folds.

Fig 18 shows the performance of the Multi-Scale CNN model across different SNR levels under five-fold cross-validation. Compared to the standard CNN, this model demonstrates

improved recognition accuracy and stability, especially in low-SNR conditions. By employing convolutional kernels of different sizes ($2 \times 2$ and $4 \times 4$), it effectively captures hierarchical features across multiple receptive fields, leading to enhanced noise robustness.

At SNR = −10 dB, the Multi-Scale CNN achieves approximately 95% accuracy, which is a significant improvement over the standard CNN at the same SNR. Furthermore, performance fluctuations across different folds are reduced, suggesting better generalization in noisy conditions.

A threshold-based analysis reveals that the model surpasses the 90% accuracy threshold starting from SNR = −12 dB, earlier than the standard CNN. This indicates that the multi-scale architecture enables the model to enter its stable performance phase under more adverse noise conditions, supporting its superior feature extraction capability.

Fig 19 highlights the performance of the Multi-Scale CNN-GRU model under varying SNR levels using five-fold cross-validation. This model integrates a GRU layer into the multi-scale convolutional framework, allowing it to capture both spatial and temporal features. As a result, it demonstrates significantly improved robustness in low-SNR conditions.

At SNR = −10 dB, the model achieves approximately 97% accuracy across all folds, and consistently reaches near-perfect accuracy for SNR <0 dB. Compared to both the standard CNN and the Multi-Scale CNN models, the CNN-GRU model offers enhanced performance stability and recognition accuracy across the entire SNR range.

The inclusion of GRU enables effective modeling of temporal dependencies in the signal, making the model more resilient to multipath fading and frequency-selective distortion, which are common in shortwave communication. In terms of threshold behavior, the MSC-GRU model is the earliest among the three to surpass the 90% accuracy mark, doing so at SNR = −14 dB, indicating its superior capacity to operate reliably in highly noisy environments.

Fig 20 provides a comparative overview of the recognition accuracy for the CNN, Multi-Scale CNN, and Multi-Scale CNN-GRU models across different SNR levels. The standard CNN model performs adequately under high SNR conditions ($\geq$ 0 dB), reaching near 100% accuracy. However, its performance degrades significantly as the SNR drops below 0 dB, indicating a lack of robustness against noise.

The Multi-Scale CNN model offers a notable improvement in low-SNR environments due to its multi-scale convolutional architecture. By capturing spatial features at multiple receptive fields, it enhances the model's ability to resist noise and variability in signal patterns. This is especially evident in the SNR range from -16 dB to -10 dB, where its accuracy remains consistently above 94%.

The Multi-Scale CNN-GRU model exhibits the best performance among all three. It maintains high accuracy across the full SNR spectrum, with minimal degradation even at SNR = −10 dB, and outperforms the other two models in both accuracy and stability.

Threshold analysis indicates that the MSC-GRU model is the first to exceed the 90% accuracy threshold, doing so at SNR = −14 dB, followed by the Multi-Scale CNN at −12 dB, and the standard CNN at −6 dB. This result reinforces the strength of combining multi-scale spatial extraction with temporal modeling in managing noise and complexity in shortwave signal recognition.

In summary, the experimental results across Figures 17 to 20 demonstrate that the recognition accuracy of all models improves with increasing SNR. However, their ability to maintain robustness under low-SNR conditions varies significantly.

The standard CNN model achieves acceptable performance only when SNR $\geq$0 dB, but its accuracy drops sharply under high-noise conditions. The Multi-Scale CNN shows noticeable improvement in noisy environments due to its multi-scale feature extraction, reaching over

90% accuracy at SNR = −12 dB. The MSC-GRU model consistently outperforms the others, demonstrating superior recognition accuracy and cross-fold stability across all SNR levels. It is the only model that exceeds 90% accuracy as early as SNR = −14 dB, proving its ability to handle complex signal distortions.

These findings validate the effectiveness of combining multi-scale spatial feature learning with temporal modeling, which is crucial for reliable shortwave protocol recognition in real-world noisy scenarios.

## Section B: Recognition of subcarriers within three shortwave protocols

The specific parameters of the network model used in Section B are consistent with those in Table 2. Since the number of classes has increased from 3 to 12, only the number of neurons in the fully connected layer has been adjusted to 12.

Fig 21 presents the recognition accuracy of the basic CNN model for subcarrier identification under varying SNR conditions, evaluated through five-fold cross-validation. As the SNR increases from −20 dB to 20 dB, the model shows a general improvement in accuracy. However, in low-SNR regions (SNR <0 dB), the recognition performance exhibits high variance across folds, with accuracy fluctuating between 40% and 80%.

This inconsistency reflects the limited robustness of the CNN model in noisy environments and suggests that its performance is highly sensitive to signal quality and fold-specific sample distributions. The lack of advanced spatial or temporal modeling restricts the model's ability to extract discriminative features in the presence of significant noise.

In terms of threshold performance, the CNN model reaches 90% accuracy only when SNR exceeds approximately 6 dB, indicating that reliable subcarrier recognition is possible only under relatively clean signal conditions.

Fig 22 depicts the subcarrier recognition accuracy of the Multi-Scale CNN model across different SNR levels, evaluated with five-fold cross-validation. Compared to the standard CNN, the Multi-Scale CNN exhibits greater stability and higher accuracy, particularly in the -10 dB to 0 dB SNR range.

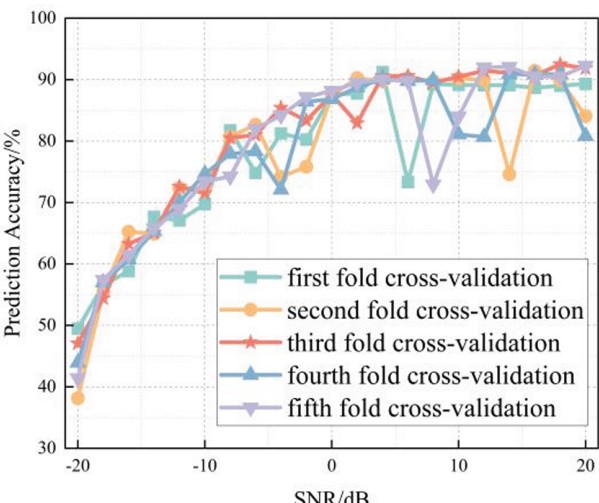

**Fig 21. Five-fold cross-validation results of CNN algorithm at different SNRs.**

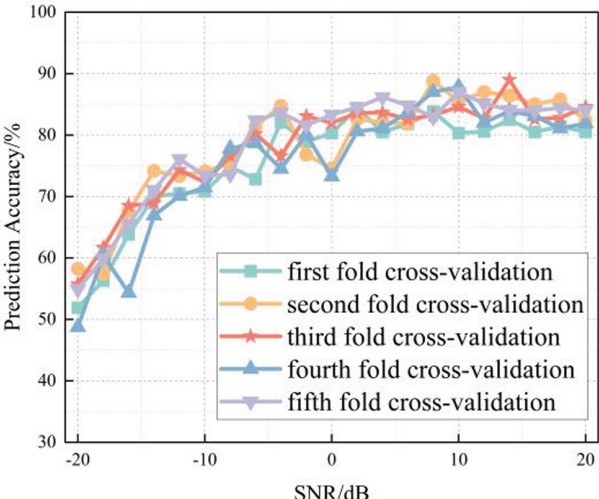

**Fig 22. Five-fold cross-validation results of mulit-scale CNN algorithm at different SNRs.**

This improvement is attributed to the use of multi-scale convolutional filters, which enable the model to extract spatial features at various receptive fields, thus improving its resistance to signal degradation caused by noise. The performance across folds also shows reduced variance, reflecting enhanced robustness in noisy environments.

From a threshold performance perspective, the Multi-Scale CNN model first exceeds 90% accuracy at around SNR = 2 dB, which is earlier than the standard CNN and indicates an improved ability to perform reliably under moderate noise.

Fig 23 shows the subcarrier recognition performance of the Multi-Scale CNN-GRU model under varying SNR levels, using five-fold cross-validation. This model consistently outperforms the other two architectures across all noise levels. In particular, at SNR <0 dB, the inclusion of the GRU layer enhances temporal dependency modeling, contributing to greater prediction stability across folds and improved robustness.

Compared to the CNN and Multi-Scale CNN models, the MSC-GRU model exhibits smoother accuracy curves and reduced inter-fold fluctuations, reflecting its superior generalization under complex signal conditions. This is especially valuable in shortwave communication environments, where signals are often degraded by multipath effects and selective fading.

In terms of threshold performance, the MSC-GRU model is the earliest to cross the 90% accuracy mark, achieving it at around SNR = 0 dB, which is significantly lower than the CNN (6 dB) and Multi-Scale CNN (2 dB), underscoring its robustness under noisy conditions.

Fig 24 provides a comparative analysis of the CNN, Multi-Scale CNN, and Multi-Scale CNN-GRU models for subcarrier modulation recognition across a range of SNR levels. Among the three, the Multi-Scale CNN-GRU model consistently achieves the highest recognition accuracy, particularly in the –10 dB to 10 dB range, where it maintains both high performance and low variance.

The standard CNN model performs poorly under low-SNR conditions, with substantial accuracy degradation. The Multi-Scale CNN provides moderate improvement through spatial feature enhancement, but lacks temporal modeling capability.

Notably, the MSC-GRU model reaches the 90% accuracy threshold earliest, at SNR ≈0 dB, whereas the Multi-Scale CNN does so at 2 dB, and the CNN only at 6 dB. These results

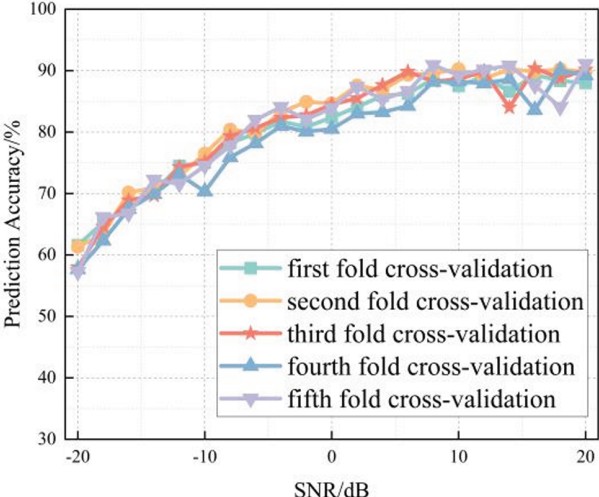

**Fig 23. Five-fold cross-validation results of mulit-scale GRU algorithm at different SNRs.**

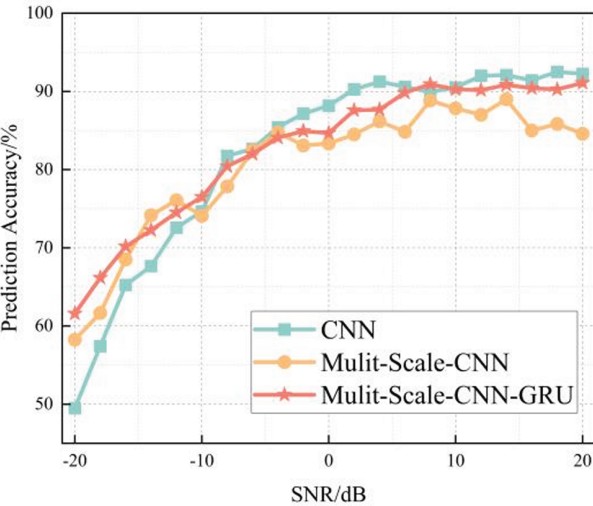

**Fig 24. Comparison of recognition rates for three algorithms at different SNRs.**

highlight  the critical advantage of combining multi-scale convolutional feature extraction with GRU-based temporal modeling, enabling the MSC-GRU model to better generalize in noisy environments common in real-world shortwave communication.

In summary, Fig 21–Fig 24 provide a comprehensive comparison of three models—CNN, Multi-Scale CNN, and Multi-Scale CNN-GRU—in the task of subcarrier modulation recognition under varying SNR levels. As the SNR increases, all models show improved accuracy, but their robustness under low-SNR conditions differs significantly.

The CNN model performs poorly in noisy environments and only achieves stable results at higher SNRs. The Multi-Scale CNN demonstrates improved accuracy and reduced fold variance by leveraging spatial feature diversity, reaching 90% accuracy at SNR = 2 dB. The Multi-Scale CNN-GRU (MSC-GRU) consistently outperforms the others, showing enhanced

temporal modeling capability and cross-fold consistency, and reaching 90% accuracy as early as SNR = 0 dB.

These results reaffirm that combining multi-scale spatial feature extraction with temporal sequence modeling (via GRU) significantly enhances recognition performance and robustness, making the MSC-GRU model highly suitable for subcarrier recognition tasks in realistic shortwave communication systems.

## Section C: Real-world evaluation under practical shortwave channels

To validate the robustness and generalization ability of the MSC-GRU model in realistic shortwave communication environments, we conducted experiments using the USRP B210 hardware platform. The experiments simulated two typical shortwave channel environments, indoor and outdoor, to explore the model's performance under different conditions.

In the outdoor environment experiment, the transmitter and receiver were set up in an open field. Here, the main impacts were free - space attenuation and natural noise sources, and the effect of multipath fading was relatively weak. As can be seen from Fig 25, the MSC-GRU model demonstrated excellent performance in the outdoor scenario. The recognition accuracy of shortwave protocols remained above 92% at all tested distances. Thanks to the relatively stable channel conditions, the model could capture signal features more effectively, showing good performance and providing strong support for its application in shortwave communication in open areas.

In the indoor environment experiment, the transmitter and receiver were placed in a space filled with reflectors (such as walls, metallic structures, and electronic equipment), resulting in severe multipath interference, non-line-of-sight propagation, and ambient electromagnetic interference. Referring to Fig 26, in the indoor environment, the recognition accuracy of the MSC-GRU model for shortwave protocols could reach above 95% at short distances (0–2 m). However, as the distance increased to 4 m, the accuracy dropped to above 87%. This indicates that the complex indoor channel environment has a certain impact on the model's performance, but the model still has a certain anti-interference ability.

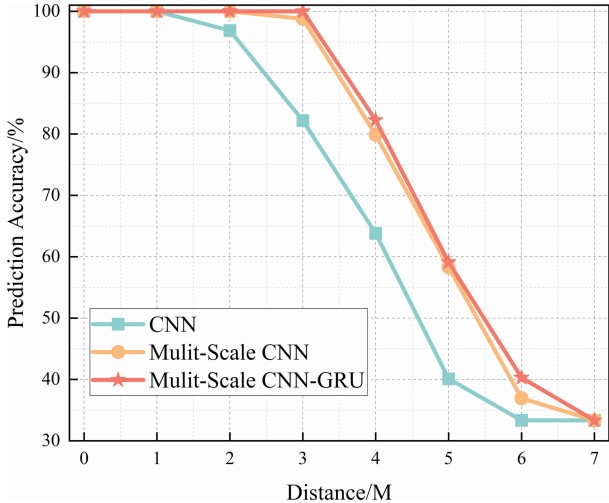

**Fig 25. Outdoor protocol recognition accuracy (%) vs. distance (m) for MSC-GRU and baseline models.**

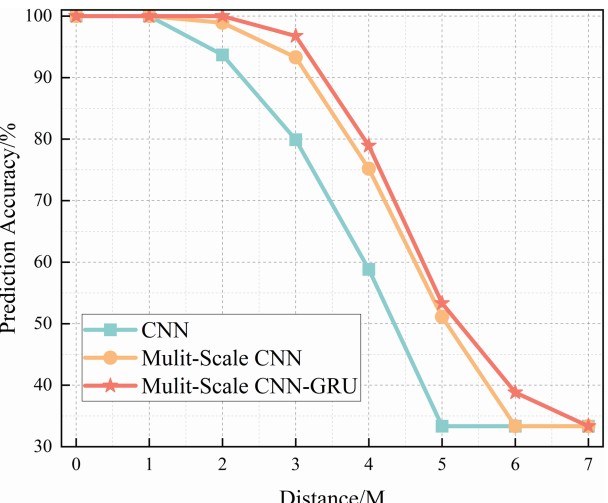

**Fig 26. Indoor protocol recognition accuracy vs. distance.**

In the outdoor subcarrier recognition experiment, as shown in Fig 27, the MSC-GRU model showed high sensitivity. The initial accuracy could reach 90.4±1.3% at 0.5 m, and it was still 82.7±1.7% at 7 m. This shows that in the outdoor environment, the model has high accuracy and stability in subcarrier recognition at different distances and can well adapt to signal changes caused by distance variations.

In the indoor subcarrier recognition experiment, as can be seen from Fig 28, the performance of the MSC-GRU model declined more significantly. The accuracy dropped from 88.2±1.% to 75.3±2.1% (with a degradation rate of 3.23%/m), and the threshold distance was only 2.3 m. This reflects that the interference of the indoor environment on signals seriously

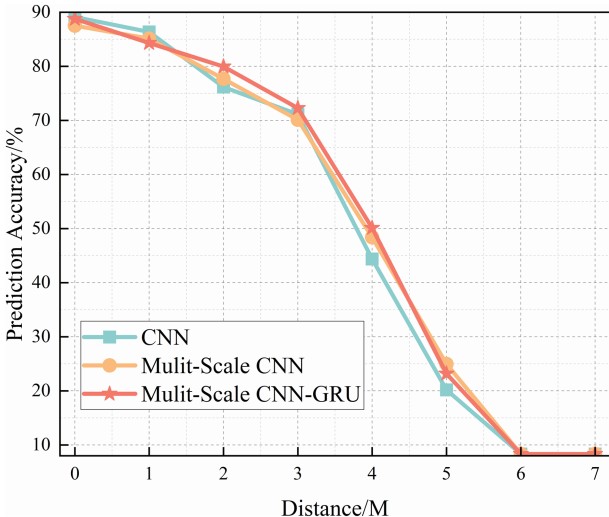

**Fig 27. Outdoor subcarrier modulation recognition accuracy vs. distance.**

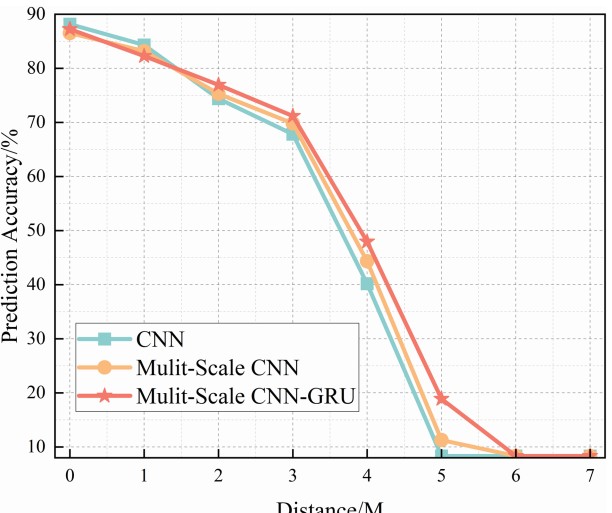

**Fig 28. Indoor subcarrier modulation recognition accuracy vs. distance.**

hinders the accurate recognition of subcarriers. Even so, the MSC-GRU model still has certain advantages over other comparative models in indoor subcarrier recognition.

Overall, the experimental results in indoor and outdoor environments show that the MSC-GRU model exhibits different performance characteristics under different shortwave channel conditions. In the outdoor environment, it shows high accuracy and stability in both short-wave protocol and subcarrier recognition. In the indoor environment, although the complex channel conditions affect the model's performance, the model can still maintain a certain accuracy in shortwave protocol and subcarrier recognition tasks, reflecting a certain anti-interference ability and generalization ability. These experimental results fully verify the feasibility and effectiveness of the MSC-GRU model in practical shortwave communication scenarios and provide important reference for its application in shortwave communication in different environments. However, the impact of the indoor environment on the model's performance also indicates that in future research, it is necessary to further optimize the model to adapt to more complex indoor shortwave communication environments.

## Conclusion

This study explores deep learning models for modulation scheme recognition in shortwave communication protocols (CLOVER-2000, 2GALE, and 3GALE) under varying SNR conditions. The models include a basic CNN, a Multi-Scale CNN, and a Multi-Scale CNN-GRU. Key findings are as follows:

1. Recognition Robustness: All models perform well at SNR ≥ 0 dB, but the Multi-Scale CNN-GRU shows the highest robustness at low SNR, especially at -10 dB, due to GRU's temporal modeling.

2. Enhanced Subcarrier Modulation Recognition: In low-SNR conditions, the Multi-Scale CNN-GRU outperforms other models, excelling at subcarrier modulation recognition by leveraging multi-scale and temporal features to manage complex signals.

3. Model Limitations: The basic CNN struggles with low-SNR noise, while the Multi-Scale CNN improves performance through multi-scale convolution but lacks strong temporal features.

4. Multi-Scale CNN-GRU Superiority: By integrating multi-scale convolution and GRU, the Multi-Scale CNN-GRU provides robust recognition, effectively handling noise and dynamic conditions in shortwave environments.

Overall, combining multi-scale convolution and GRU significantly enhances recognition accuracy, particularly in challenging low-SNR scenarios, supporting its potential for shortwave communication applications.

## Author contributions

**Conceptualization:** Jiuxiao Cao, Rui Zhu, Zhen Wang.

**Data curation:** Jiuxiao Cao, Rui Zhu, Zhen Wang, Jun Wang, Guohao Shi.

**Formal analysis:** Jiuxiao Cao, Rui Zhu, Zhen Wang, Peng Chu.

**Funding acquisition:** Rui Zhu, Zhen Wang, Peng Chu.

**Investigation:** Rui Zhu.

**Methodology:** Jiuxiao Cao, Rui Zhu, Zhen Wang, Jun Wang, Guohao Shi.

**Project administration:** Jiuxiao Cao, Rui Zhu.

**Resources:** Jiuxiao Cao, Rui Zhu, Jun Wang.

**Software:** Jiuxiao Cao, Rui Zhu, Zhen Wang, Jun Wang, Guohao Shi.

**Supervision:** Jiuxiao Cao, Rui Zhu, Zhen Wang, Jun Wang, Guohao Shi.

**Validation:** Jiuxiao Cao, Rui Zhu, Zhen Wang, Jun Wang, Guohao Shi, Peng Chu.

**Visualization:** Jiuxiao Cao, Rui Zhu, Zhen Wang, Jun Wang, Guohao Shi, Peng Chu.

**Writing – original draft:** Jiuxiao Cao.

**Writing – review & editing:** Jiuxiao Cao.

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
