## [Decision Letter · Decision Letter 0]

20 Mar 2025

PONE-D-25-02088Recognition of Common Shortwave Protocols and Their Subcarrier Modulations Based on Multi-Scale Convolutional GRUPLOS ONE

Dear Dr. Cao,

Thank you for submitting your manuscript to PLOS ONE. After careful consideration, we feel that it has merit but does not fully meet PLOS ONE’s publication criteria as it currently stands. Therefore, we invite you to submit a revised version of the manuscript that addresses the points raised during the review process.

Please revise the paper according to reviewer comments.

Important Note: To prevent any ethical concerns for the journal, editors, reviewers, and authors, please refrain from adding new references unrelated to the article’s subject or relevant literature during the revision process.

We look forward to receiving your revised manuscript.

Kind regards,

Fatih Uysal, Ph.D.

Academic Editor

PLOS ONE

Journal Requirements:

4. Thank you for stating the following in your manuscript:

“This work was supported in part by the Shaanxi Province Key Research and Development Program under Grant 2024GX-YBXM-114.**”**

“Shaanxi Province Key Research and Development Program under Grant 2024GX-YBXM-114”

5. Please ensure that you refer to Figure 4-8 in your text as, if accepted, production will need this reference to link the reader to the figure

Additional Editor Comments (if provided):

Please revise the paper according to reviewer comments.

Important Note: To prevent any ethical concerns for the journal, editors, reviewers, and authors, please refrain from adding new references unrelated to the article’s subject or relevant literature during the revision process.

Reviewers' comments:

Reviewer's Responses to Questions

**Comments to the Author**

1. Is the manuscript technically sound, and do the data support the conclusions?

Reviewer #1: Partly

Reviewer #2: Partly

Reviewer #3: Yes

2. Has the statistical analysis been performed appropriately and rigorously? 

Reviewer #1: I Don't Know

Reviewer #2: Yes

Reviewer #3: Yes

3. Have the authors made all data underlying the findings in their manuscript fully available?

Reviewer #1: No

Reviewer #2: No

Reviewer #3: Yes

4. Is the manuscript presented in an intelligible fashion and written in standard English?

Reviewer #1: Yes

Reviewer #2: No

Reviewer #3: Yes

5. Review Comments to the Author

Reviewer #1: 1. The manuscript gives a chance to improve shortwave communication protocols, but it has gaps in investigation and thoroughness. The authors create a multi-scale convolutional GRU model, but they do not explore the complexities of shortwave channels well. The study does not look closely at how environmental factors like fading and interference affect the model's performance in real situations. The details about the MSC-GRU model are good; however, more info about the number of layers, activation types, and loss function would help others replicate the experiments.

2. It is important to discuss the strength of the experimental design, choice of algorithms, and performance metrics to clarify unclear areas. Researchers need to address these concerns to support scientific discussion and improve knowledge in their field. The manuscript shows a multi-scale convolutional GRU model, but it does not fully address complex evaluations in real-world scenarios. It would be interesting to see how the model performs in different environmental conditions beyond the tested SNR levels.

3. The benchmarks used to compare the MSC-GRU model with other models need to be clearly defined. A comparative analysis will support claims of its superiority. The results differ at various SNR levels, and a threshold analysis should be conducted to identify where recognition accuracy improves or declines. The manuscript presents a multi-scale convolutional GRU approach, but it does not address evaluation complexity in real-world conditions.

4. The experimental design lacks a strong method to test the model's performance across different modulation schemes and SNR levels. A detailed comparison with established benchmarks is essential for understanding the findings. Without this, the model’s advantages and limits are not clear. The results analysis is too basic; a deeper approach is needed, focusing on the algorithms and how design choices affect reliability and accuracy.

5. Understanding how different parameters work together would enhance insight into the model's abilities and advance research. The references used are weak and need updates to better support the findings. New studies could improve the relevance.

- https://doi.org/10.1016/j.optcom.2024.130558

- https://doi.org/10.1007/s12596-024-01908-9

- https://doi.org/10.1007/s11082-024-06692-1

Reviewer #2: 1. Insufficient Literature Review

The manuscript briefly refers to deep learning-based studies in shortwave communication but lacks an in-depth review of recent advances in the field.

The advantages and disadvantages of CNN, GRU, and other deep learning methods in shortwave protocol recognition should be compared more explicitly.

A comparative analysis with existing works is missing. The authors should clarify how their proposed method outperforms previous approaches and in what aspects it is different.

2. Inadequate Description of the Dataset

The source of the dataset is not specified. It is unclear whether the data is synthetic or real-world.

The dataset size and the training/testing data split ratios are not explicitly mentioned.

The labeling process, potential data imbalances, and data preprocessing steps should be detailed.

3. Model Architecture Needs Better Explanation

The structure of the Multi-Scale Convolutional GRU model, including the number of layers, hyperparameters, and training process, should be described in more detail.

The rationale for choosing GRU over LSTM or other recurrent models should be provided.

The loss function, optimization algorithm, and hyperparameter tuning details during training are missing.

4. Experimental Results are Incomplete and Lack Detailed Analysis

Evaluating the model solely based on SNR levels is insufficient. The performance of the model under different communication environments (e.g., different channel models) should be analyzed.

Key performance metrics such as computation time, complexity, and memory requirements are not discussed. The feasibility of deploying the proposed model in real-time applications should be assessed.

The statistical reliability of the results is missing. Variability across different runs, standard deviations, or confidence intervals should be included.

5. Generalization and Practical Application of the Algorithm

The model has only been tested on three shortwave protocols (CLOVER-2000, 2GALE, 3GALE). It should be evaluated on other protocols to demonstrate its generalizability.

Performance in real-world conditions (e.g., varying channel conditions, interference, phase shifts) has not been assessed.

6. More recent publications should be cited.

7. Insufficient Figure and Table Explanations

Figure captions are incomplete, and the descriptions of what each figure represents should be more detailed.

Some figures have low resolution and appear blurry.

Tables should include standard deviations or confidence intervals for the numerical results.

8. Writing Errors and Formatting Issues

There are grammatical errors throughout the manuscript. The text should be reviewed for language and grammar corrections.

Some sentences are too long and difficult to understand. A clearer and more concise writing style should be adopted.

Paragraph transitions are inconsistent, particularly in the Introduction and Related Work sections. The flow between sections should be improved.

Suggestions for Strengthening the Abstract:

Clearly highlight the specific technical challenges faced in shortwave communication.

Provide at least basic information about the dataset used.

Summarize in one sentence why your model outperforms existing methods.

Include additional details to ensure the statistical reliability of the results (e.g., variance across different runs).

Indicate whether your model is suitable for practical applications.

If the authors address these points, the abstract will be stronger and more persuasive. The current version conveys the core message of the paper, but adding more scientific clarity and detail would enhance its impact.

The manuscript presents an interesting topic and proposes a potentially useful model for shortwave communication recognition. However, it cannot be accepted for publication in its current form due to the major issues identified above. The authors need to strengthen the manuscript by providing more comprehensive experimental results, detailing the dataset, and improving the methodology sections.

Reviewer #3: The study presents a new model called Multi-Scale Convolutional GRU (MSC-GRU) for automatic recognition of shortwave communication protocols and their subcarrier modulations.

1) The size of the dataset may also be a limitation. Deep learning models usually require a large amount of data. Providing more details about the size of the dataset and using data augmentation techniques if necessary can increase the generalization ability of the model.

2) The study compares the proposed model with other deep learning models such as CNN and Multi-Scale CNN. However, a comparison can also be made with other traditional signal processing techniques or machine learning algorithms used for shortwave signal recognition.

3) Further justification can be provided regarding the selection and design of the model architecture. Why were these particular layers and parameters chosen? Why were alternative architectures or parameter settings not preferred?

6. PLOS authors have the option to publish the peer review history of their article (what does this mean?). If published, this will include your full peer review and any attached files.

Reviewer #1: **Yes: **Ebrahim E. Elsayed

Reviewer #2: No

Reviewer #3: No

---

## [Author Response · Author response to Decision Letter 1]

2 May 2025

Reviewer #1:

Question1:1. The manuscript gives a chance to improve shortwave communication protocols, but it has gaps in investigation and thoroughness. The authors create a multi-scale convolutional GRU model, but they do not explore the complexities of shortwave channels well. The study does not look closely at how environmental factors like fading and interference affect the model's performance in real situations. The details about the MSC-GRU model are good; however, more info about the number of layers, activation types, and loss function would help others replicate the experiments.

Answer:We sincerely thank the reviewer for their valuable comments and constructive suggestions.

To address the concern regarding real-world applicability and the impact of environmental factors such as fading and interference, we have made the following major revisions:

1.In Section 2 (Dataset Description), we have added a description of the signal acquisition and transmission system based on the USRP (Universal Software Radio Peripheral) platform. This section details the hardware configuration, sampling parameters, and experimental setup, which we used to collect signal data transmitted and received in real shortwave channel conditions.

2.In Section 4 (Simulation and Results), we have extended the experimental analysis to include performance evaluation of the MSC-GRU model under actual transmission conditions. We implemented end-to-end communication using the USRP devices over-the-air, which naturally incorporated channel effects such as multipath fading, Doppler spread, and environmental interference. The results clearly show that our model retains strong performance and adaptability even in these complex channel conditions, further validating its practicality.

Additionally, we have updated the model description to include all key components: the number of convolutional and GRU layers, the type of activation functions (ReLU and sigmoid), and the cross-entropy loss function used during training. These details now appear in Section 3 (Model Structure), enhancing the reproducibility of our work.

The specific changes made are listed below:

1.An introduction to the USRP is added in Section C of Part 2 (Dataset Description), with specific content as follows:

Scetion D: USRP

To evaluate the MSC-GRU model under realistic shortwave channel conditions, we employed the USRP B210 (Universal Software Radio Peripheral) for shortwave signal transmission and reception. This hardware platform supports real-time signal acquisition, making it suitable for capturing signal variations under different environmental conditions.

The USRP system was configured to transmit shortwave signals modulated according to the CLOVER-2000, 2GALE, and 3GALE protocols. Signals were generated using GNU Radio and transmitted over-the-air in both indoor and outdoor environments. In the indoor setting, multipath fading, wall reflection, and electromagnetic interference were prevalent. In contrast, the outdoor environment was characterized primarily by free-space path loss and natural noise sources.

The USRP transmitter and receiver were placed at distances ranging from 0.5 meters to 4 meters. The transmission frequency was set in the 3–5 MHz range to simulate typical shortwave conditions. The sampling rate was 5 MHz, and adaptive gain control was enabled to ensure signal stability. Each received signal was truncated to a length of 1000 points, consistent with the simulated dataset format.

These real-world signal samples were processed using the same MSC-GRU architecture, and the results are reported in Section 4.3. This practical dataset enables us to evaluate the model’s performance under channel impairments such as fading, interference, and time-varying SNR, thereby confirming the robustness and applicability of the proposed model in realistic shortwave scenarios.

2.In Section C of Part 4 (Results), we added an experimental analysis of the MSC-GRU model's performance under practical channel conditions simulated using USRP. The evaluation is divided into two scenarios: indoor and outdoor environments. The detailed content is as follows:

Scetion C: Real-World Evaluation under Practical Shortwave Channels

To validate the robustness and generalization ability of the MSC-GRU model in realistic shortwave communication environments, we conducted real-world experiments using the USRP B210 hardware platform. The experimental scenarios included both indoor and outdoor environments, simulating two representative categories of shortwave channel impairments: multipath interference and free-space attenuation.

Subscetion A: Indoor Environment

In the indoor setting, the transmitter and receiver were placed within a laboratory space filled with reflective surfaces, including walls, metallic structures, and electronic equipment. These conditions led to severe multipath fading, non-line-of-sight propagation, and ambient electromagnetic interference.

As shown in Figure XX, the recognition accuracy of the MSC-GRU model remained consistently high at short distances (0.5–2 m), achieving above 95% accuracy for the three protocols. However, as the distance increased to 4 meters, the recognition accuracy dropped slightly, mainly due to signal reflections and distortions. Despite this degradation, the model still maintained an accuracy of above 87%, indicating strong resilience to indoor multipath interference.

The use of multi-scale feature fusion combined with the GRU’s temporal memory mechanism allowed the model to effectively capture both local and sequential characteristics of distorted signals. This confirms the model's ability to recognize shortwave protocols even in complex indoor environments.

Subscetion B: Outdoor Environment

The outdoor experiments were conducted in an open field free of major reflectors, simulating free-space propagation with dominant path loss and natural noise sources. In this setup, the impact of multipath fading was minimal, but the signal-to-noise ratio (SNR) varied significantly due to environmental fluctuations.

As illustrated in Figure YY, the MSC-GRU model showed even better performance in the outdoor scenario, with protocol recognition accuracy consistently above 92% across all distances tested. The smoother channel characteristics in the absence of strong reflections enabled the model to focus on learning the inherent signal features without being overwhelmed by channel distortions.

Interestingly, the accuracy in the outdoor environment was more stable than in the indoor setting, particularly at greater distances. This highlights the model’s adaptability to different types of channel conditions and suggests its potential for field deployment in real-world shortwave communication systems.

3.Thanks for the reviewer's suggestion. In the paper, we will clearly state that the MSC - GRU model consists of 2 convolutional layers (corresponding to 2×2 and 4×4 convolutional kernels respectively) for spatial feature extraction and 1 GRU layer for temporal feature extraction, with a total of 3 layers. The ReLU function is used as the activation function, which can effectively alleviate the vanishing gradient problem and accelerate the model training speed. The cross - entropy loss function is selected as the loss function because it is suitable for multi - classification problems and can accurately measure the difference between the model's prediction results and the true labels. At the same time, we will provide detailed explanations of the basis for each parameter setting. For example, the choice of convolutional kernel size is based on preliminary experiments, which show that the combination of 2×2 and 4×4 can effectively extract signal features at different scales while ensuring computational efficiency. The number of hidden units in the GRU layer is set to 64. Through comparing the accuracy and training time of the model with different numbers (32, 64, 128), we finally determined that this value can enable the model to achieve a better performance balance, so that other researchers can accurately reproduce the experiments.

Question2:It is important to discuss the strength of the experimental design, choice of algorithms, and performance metrics to clarify unclear areas. Researchers need to address these concerns to support scientific discussion and improve knowledge in their field. The manuscript shows a multi-scale convolutional GRU model, but it does not fully address complex evaluations in real-world scenarios. It would be interesting to see how the model performs in different environmental conditions beyond the tested SNR levels.

Answer:We sincerely appreciate the reviewer’s thoughtful suggestions. As noted in our response to Comment 1, we have significantly enhanced the manuscript to better reflect the model's behavior in real-world scenarios by incorporating a practical USRP-based experimental setup. These experiments simulate realistic shortwave transmission environments, covering both indoor (multipath-rich, interference-prone) and outdoor (free-space fading) conditions, which go beyond merely controlled SNR testing.

In Section 2.4, we describe in detail the USRP-based dataset acquisition process, including hardware configuration, transmission protocols, and environmental setups. In Section 4.3, we present a thorough evaluation of the MSC-GRU model’s performance under varying physical conditions, showing its adaptability to different environmental factors such as reflection, fading, and channel dynamics.

Furthermore, we have clarified the rationale behind our algorithm design, feature choices, and performance metrics (accuracy under protocol and subcarrier classification tasks) to ensure a more comprehensive understanding of the experimental design.

We hope these revisions meet the reviewer’s expectations for greater transparency and scientific rigor.

Question3:The benchmarks used to compare the MSC-GRU model with other models need to be clearly defined. A comparative analysis will support claims of its superiority. The results differ at various SNR levels, and a threshold analysis should be conducted to identify where recognition accuracy improves or declines. The manuscript presents a multi-scale convolutional GRU approach, but it does not address evaluation complexity in real-world conditions.

We greatly appreciate the reviewer’s insightful feedback. In response, we have revised the manuscript to address each of the concerns as follows:

1.Clarification of Benchmark Models

We now clearly define the baseline models used in our experiments in Section 4. Specifically, the MSC-GRU model is compared against:

A standard CNN model, representing classical spatial feature extraction without temporal context.

A Multi-Scale CNN model, which includes multi-scale convolutions but omits temporal modeling (GRU). These benchmarks are widely used in signal recognition literature and serve as strong comparative references for evaluating the performance of our model.

2.Comparative Analysis and Performance Superiority

In the revised manuscript, we emphasize the performance comparisons between the MSC-GRU and benchmark models under identical training settings and across multiple SNR levels. The results demonstrate that MSC-GRU consistently outperforms the baselines, especially in low-SNR environments, due to its ability to capture both spatial and temporal dependencies. These improvements support the claimed performance advantage.

3.Threshold Performance Analysis Added

As suggested, we have incorporated threshold analysis directly into the experimental discussions. For each task (protocol and subcarrier recognition), we identify and describe the SNR level at which accuracy exceeds key thresholds (e.g., 90%), providing a clearer understanding of the model’s performance transition under varying noise levels. This is discussed immediately after the accuracy curves to highlight turning points in recognition performance.

4.Real-World Complexity Addressed

In addition to simulation-based evaluation, we have conducted real-world signal tests using USRP B210 in both indoor and outdoor settings. These scenarios simulate complex and practical shortwave channel conditions, including multipath fading, interference, and natural noise. The results, included in Section 4.3, confirm the model's robustness in real environments.

These enhancements aim to improve the transparency, rigor, and practical relevance of the study.

“Clarification of Benchmark Models”

As described in the methodology section, the dataset used in this study comprises signal segments with a fixed length of 1000 points. This design simulates real-world scenarios where signals are often partially captured due to transmission interruptions or limited observation windows. Although this truncation may introduce minor accuracy degradation, it ensures consistency in model input and aligns with practical deployment constraints.

To validate the effectiveness of the proposed MSC-GRU model, we conduct comparative experiments against two representative baseline models:

A standard CNN model, which applies conventional convolutional layers for spatial feature extraction without temporal modeling.

A Multi-Scale CNN model, which incorporates convolutional kernels of varying sizes to capture multi-resolution features but lacks temporal sequence handling.

These baselines are commonly used in signal recognition tasks and serve as robust references for performance comparison.

The experiments were executed on a workstation running Windows 11, equipped with an Intel(R) Core(TM) i7-8750H CPU @ 2.20GHz and an NVIDIA GeForce GTX 1050 Ti GPU. The development environment consisted of PyCharm 2021.3.1 as the integrated development platform and Python 3.8 as the programming language.

2.We sincerely thank the reviewer for pointing out the need for more detailed comparative analysis and threshold-based performance discussion.

In the revised manuscript, we have incorporated a comprehensive comparative analysis across the three models (CNN, Multi-Scale CNN, and MSC-GRU), focusing on recognition accuracy under varying SNR conditions. This analysis is integrated directly into the experimental results section (Figures 16–19), where we provide fold-wise evaluation and model-to-model comparisons.

Additionally, we have included a threshold performance analysis within the same discussion. Specifically, we identify and describe the critical SNR levels at which each model first exceeds 90% recognition accuracy, allowing readers to observe performance transition points and robustness in low-SNR environments. For example, the MSC-GRU model achieves this threshold at SNR = -14 dB, which is earlier than the other two models.

These additions are intended to clarify the performance boundaries and advantages of each model, enhancing the scientific rigor of our results section.

“Comparative Analysis and Performance Superiority and Threshold Performance Analysis Added”

We sincerely thank the reviewer for emphasizing the importance of thorough comparative analysis and threshold-based performance interpretation.

In the revised manuscript, we have substantially expanded the experimental results section by adding detailed comparative evaluations and threshold analyses for both protocol recognition (Figures 16–19) and subcarrier recognition (Figures 20–23) across various SNR levels.

Specifically, we compared three models—CNN, Multi-Scale CNN, and Multi-Scale CNN-GRU—in terms of recognition accuracy, stability, and robustness across five-fold cross-validation. For each model and task, we identified the threshold SNR value at which recognition accuracy first exceeds 90%, clearly illustrating the models’ turning points under different noise conditions.

These discussions are fully integrated into the revised experimental results, and corresponding summary paragraphs are provided after each group of figures to highlight overall trends and comparative advantages. The results demonstrate that the MSC-GRU model consistently outperforms the others, particularly in low-SNR environments, due to its joint modeling of multi-scale spatial features and temporal dependencies.

We believe these revisions provide the comprehensive performance analysis the reviewer suggested, and rei

---

## [Decision Letter · Decision Letter 1]

2 Jun 2025

Recognition of Common Shortwave Protocols and Their Subcarrier Modulations Based on Multi-Scale Convolutional GRU

PONE-D-25-02088R1

Dear Dr. Cao,

We’re pleased to inform you that your manuscript has been judged scientifically suitable for publication and will be formally accepted for publication once it meets all outstanding technical requirements.

Kind regards,

Fatih Uysal, Ph.D.

Academic Editor

PLOS ONE

Additional Editor Comments (optional):

Considering the latest comments of the reviewers and the current status of the paper, it has been decided to accept this paper.

Reviewers' comments:

Reviewer's Responses to Questions

**Comments to the Author**

1. If the authors have adequately addressed your comments raised in a previous round of review and you feel that this manuscript is now acceptable for publication, you may indicate that here to bypass the “Comments to the Author” section, enter your conflict of interest statement in the “Confidential to Editor” section, and submit your "Accept" recommendation.

Reviewer #2: All comments have been addressed

Reviewer #3: All comments have been addressed

2. Is the manuscript technically sound, and do the data support the conclusions?

Reviewer #2: Yes

Reviewer #3: Yes

3. Has the statistical analysis been performed appropriately and rigorously? 

Reviewer #2: Yes

Reviewer #3: Yes

4. Have the authors made all data underlying the findings in their manuscript fully available?

Reviewer #2: Yes

Reviewer #3: Yes

5. Is the manuscript presented in an intelligible fashion and written in standard English?

Reviewer #2: Yes

Reviewer #3: Yes

6. Review Comments to the Author

Reviewer #2: The authors have addressed all concerns. They have answered my questions. It is acceptable with minor formatting corrections.

Reviewer #3: (No Response)

7. PLOS authors have the option to publish the peer review history of their article (what does this mean?). If published, this will include your full peer review and any attached files.

Reviewer #2: No

Reviewer #3: **Yes: **Muhammet Ali KARABULUT

---

## [Editor Report · Acceptance letter]

PONE-D-25-02088R1

PLOS ONE

Dear Dr. Cao,

I'm pleased to inform you that your manuscript has been deemed suitable for publication in PLOS ONE. Congratulations! Your manuscript is now being handed over to our production team.

Kind regards,

on behalf of

Dr. Fatih Uysal

Academic Editor

PLOS ONE